# Recent Advances in Video Analytics for Rail Network Surveillance for Security, Trespass and Suicide Prevention—A Survey

**DOI:** 10.3390/s22124324

**Published:** 2022-06-07

**Authors:** Tianhao Zhang, Waqas Aftab, Lyudmila Mihaylova, Christian Langran-Wheeler, Samuel Rigby, David Fletcher, Steve Maddock, Garry Bosworth

**Affiliations:** 1The University of Sheffield, Sheffield S1 3JD, UK; tzhang81@sheffield.ac.uk (T.Z.); waqasaftabmalik@gmail.com (W.A.); clangran-wheeler1@sheffield.ac.uk (C.L.-W.); sam.rigby@sheffield.ac.uk (S.R.); d.i.fletcher@sheffield.ac.uk (D.F.); s.maddock@sheffield.ac.uk (S.M.); 2Network Rail, Milton Keynes MK9 1EN, UK; garry.bosworth@networkrail.co.uk

**Keywords:** surveillance, rail network systems, image and video analytics, computer vision, machine learning, sensors, video anomaly detection

## Abstract

Railway networks systems are by design open and accessible to people, but this presents challenges in the prevention of events such as terrorism, trespass, and suicide fatalities. With the rapid advancement of machine learning, numerous computer vision methods have been developed in closed-circuit television (CCTV) surveillance systems for the purposes of managing public spaces. These methods are built based on multiple types of sensors and are designed to automatically detect static objects and unexpected events, monitor people, and prevent potential dangers. This survey focuses on recently developed CCTV surveillance methods for rail networks, discusses the challenges they face, their advantages and disadvantages and a vision for future railway surveillance systems. State-of-the-art methods for object detection and behaviour recognition applied to rail network surveillance systems are introduced, and the ethics of handling personal data and the use of automated systems are also considered.

## 1. Introduction

Rail network systems are, by design, open, accessible and welcoming environments. Control of entry is largely on the basis of revenue protection, not security, and with a few exceptions (e.g., international services), they do not share the strict perimeter security model adopted for air travel. While this is entirely appropriate for efficient transport operation, it does leave the systems vulnerable in several ways. Terrorist activity is often identified as the highest-profile vulnerability, and a range of resilience and design strategies have been adopted to reduce the attractiveness of railways as a target and to enable them to recover rapidly if attacked [1]. However, disruptions are caused by unexpected events such as trespass (often linked with theft of materials or equipment) and suicide attempts [2,3]. This requires that railway employees are trained to look out for and offer support to people who may be considering taking their own life on the railway.

Mitigation measures such as fencing can be used to prevent access at high-risk locations and help address security, trespass and suicide risks. However, access prevention is not a solution for station environments, and it is here that recent advances in video analytics tailored to rail [4,5] offer alternative approaches to reduce these risks in coordination with ‘designing in’ physical solutions such as blast resistant glazing [6], and management solutions such as staff training to identify out-of-the-ordinary situations. The development of digital technology for sensors, sensing, data fusion and machine learning, combined with the advancing communication and cloud technologies have the potential to contribute greatly in these areas. Their application, however, requires careful assessment of ethics concerning public privacy and data handling, for example, as regulated in Europe through General Data Protection Regulation (GDPR) regulations. This survey supports the application of these technologies in rail by bringing together technical and ethical information to protect public security, protect the rail system from trespassing, and reduce railway suicides. This range of applications is useful in correctly balancing the benefits and costs of a potential system, but it leads to difficulty in the choice of language. Where video analytics of a potential attacker may aim to identify a ‘target’ for investigation, the same technology could also be identifying a vulnerable person in distress. For simplicity, in this article, we have adopted the most common language used in each area of technology.

The main contributions of this work are the following: (i) it provides a thorough survey of the latest trends in closed-circuit television (CCTV) systems, with their advantages and disadvantages, (ii) presents the latest trends especially for vision-based behaviour analytics and especially in the light of improving safety and security in rail network stations, (iii) presents a systematic approach encompassing the whole process from the data collection, to storage, communications and video analytics, including ethics and data protection aspects, (iv) traces a vision for the future autonomous vision-based surveillance on several aspects, including the impact of the uncertainties in Artificial Intelligence models.

This paper is organised in the following way. Section 2 introduces an overview of current surveillance systems in rail networks. Section 2.1 introduces the recent research trends in railway video surveillance. Following by the main components of current surveillance systems, including CCTV cameras and non-conventional sensors, and a brief overview of monitoring and recording systems and various algorithms on anomaly detection in surveillance video data. The strengths of multiple modality sensors are discussed to help in sensor selection concerning the level of required security. Section 2.5.1 introduces the modern deep learning methods used in video analytics. Various computer vision technologies built with Artificial Intelligence (AI) models based on deep learning algorithms used in railway network surveillance are introduced in Section 2.1.1.

The impact of the uncertainties in computer vision technologies and state-of-the-art Bayesian deep learning methods for formulating the uncertainties are introduced in Section 3. A broad vision of next-generation systems is given in Section 4, and future data centres are introduced in Section 4.1. Ethical and privacy-related aspects are covered in Section 5, followed by conclusions in Section 6.

## 2. Overview of Surveillance Systems for Rail Networks

CCTV systems [7] have evolved over the years from simple analogue systems in the 1960s to digital systems developed in the 1980s [8]. The advent of digital systems extended the limited recording and distribution abilities of analogue systems. Object detection and tracking are the basic video analytics technologies in digital systems. Since the 2000s, more and more advanced video analytics technologies have been developed, taking advantage of valuable video data collected by multiple integrated sensors.

### 2.1. Bibliometric Analysis

Nowadays, the most advanced video analytics for CCTV is capable of providing automated detection and alerts based on information processing, reasonable framework and computer vision technologies such as object detection [9], behaviour analysis and activity recognition [10,11,12], crowd tracking [13], emergency evacuation and deterrence [14,15,16,17]. We have analysed the current trends in CCTV video surveillance for rail network systems and related computer vision technologies. Some of the results since 2010 are shown in Figure 1 and Figure 2. The following keywords are used in the search: “surveillance”, “railway”, “computer vision” and “machine learning”. A tool called Bibliometrix [18] is used here.

During the construction of surveillance systems of railway networks, there are certain rules issued by the government or relevant authorities that must be followed, which are known as requirements documents. Requirements documents related to the standards for surveillance systems in different countries [19,20,21,22,23,24,25,26,27,28,29] are compulsory to follow and are to be prepared based on the outcome of the previous step. It specifies the needs and the justification of the CCTV system [30]. Additionally, these documents cover the legislative and privacy limitations. Other kinds of requirements documents include multiple instructions on building the surveillance systems, for example, how to access data sets on video analysis [31]. An example of legislative limitation is to assume that the recording capacity of the system is one year, whereas the storage time limit defined by the government is actually six months. An example of privacy limitations is blocking the view of a private neighbouring property from the system.

#### 2.1.1. Trends in Computer Vision Technologies in Video Surveillance

These technologies are either currently used in surveillance systems in railway networks or considered essential to be developed in the next generation of smart surveillance systems. Table 1 shows the recent research status of the main computer vision technologies used in railway video surveillance. The result is concluded under the keywords: “surveillance”, “railway”, and “the name of computer vision technology” using the Google Scholar search tool. These statistical results show that the focus is on human activity recognition, anomaly detection and crowd analysis.

Section 2.2 describes the main objectives of CCTV surveillance systems.

### 2.2. Objectives of CCTV Surveillance Systems

The primary objective of the CCTV systems is to assist in preventing crime and terrorist activities and to provide safety and security to people. Sensors are compulsory elements of the CCTV system. Conventional video-based sensors have limited capability of identifying hazardous materials or weapons used by terrorists or criminals. The main components for a surveillance system are various sensors, monitoring and recording systems, video analytics, communication networks, cloud computing or other storage systems. Communication networks are critical for ensuring the reliable transfer of data among these components. The sensors, video analytics and recording systems are configured from the monitoring system. A block diagram of a CCTV system is shown in Figure 3.

Raw sensor data and the data analytics output can overwhelm security operators if the data are inefficient. In order to detect behaviours of interest correctly, carefully designing an efficient monitoring system is important. A satisfying surveillance system setting relies on the integration of the geographic information system such as the Global Positioning System (GPS) with the CCTV to reference information from a known location. This reduces the training time of the operators as effective monitoring does not solely rely on their knowledge of sensor placement across the site. Data from multiple sensors is combined to provide all the information and intelligence relevant to a specific area in one place. Advanced display software provides an option to view the scene not only from the camera’s point of view but also from multiple viewpoints, which can enhance situational awareness and reduce response time.

Taking the advantages of the surveillance systems built in the railway networks, abnormal behaviour detection methodologies by computer vision methods are widely implemented. Surveillance systems built for railway networks are usually focused on a constrained area, which means there are multiple computer vision technologies that could be applied to specific scenarios. These anomaly detection algorithms based on machine learning methods can autonomously detect unusual behaviours in video surveillance data through the camera sensors without operator intervention. By detecting the change of movement temporally and spatially, the system should be intelligent enough to classify behaviours as regular or abnormal events [8,32,33,34]. State-of-the-art methods for anomaly detection in video surveillance are presented in Section 2.1.1.

### 2.3. Sensors

The most commonly used sensors for collecting data are optical and infrared (IR) cameras, chemical sensors and motion detectors. CCTV cameras are an important element of security and surveillance systems [7] when deployed as part of a multi-layered security architecture and provide video streams for all areas, both indoor and outdoor.

Conventional sensors, such as video cameras providing behavioural trait analysis, rely on advances in video analysis to detect the behaviour of interest. Non-conventional sensors, e.g., chemical sensors, have their advantages in detection through analysis of vapours, with limitations in tracking and data visualisation. The integration of non-conventional systems is still promising in current and future systems. Other combined conventional and non-conventional sensors, such as health sensors that are useful for detecting and preventing viruses and infections, have also been of benefit during the COVID-19 pandemic. Train stations are major entrances and exits in national and international transport networks. With these kinds of sensors, the surveillance systems could aid in the population-level understanding of epidemic or pandemic spread.

Multimodality (conventional and non-conventional) sensors collaborate to give early warnings and share useful data for sensor management purposes. Data from a single sensor is considered low-level intelligence, whereas automated analysis of the sensor’s data provides higher level intelligence to the security operators. Figure 4 illustrates such a multi-modal sensor framework.

With the continuous advancement of human technology, smart phones or watches carried by passengers have been developed as other types of modern sensors to collect behaviour data for behaviour surveillance [35]. However, due to the privacy and security restrictions, these types of sensors have not been able to be used in surveillance systems.

#### 2.3.1. Cameras

The most important component of a CCTV system is a camera, which is also known as an imaging sensor. In some systems, non-imaging sensors are integrated to enhance operations. The two common imaging sensors in the CCTV system are optical and thermal cameras. The cameras consist of three parts: lens, image sensor and image processor. The lens focuses the light reflected from an object on the image sensor, the image sensor converts the light photons to electrical signals, and the image processor organizes, optimizes and transmits the electrical signals. Some advanced cameras have onboard video analytics as well. The cameras are available in various options. A summary of basic and value-added options is given in Table 2 and Table 3. These tables can be referred to for choosing the camera features as per the risk assessment.

#### 2.3.2. Chemical Sensors

Camera sensors are not capable of detecting potential threats in some cases, such as explosive materials hidden in bags. Chemical sensors are another type of sensor commonly deployed in public transport stations to detect hazardous materials. Chemical sensors detect objects by sensing the vapour concentration leaked from the radioactive and explosive materials. The localisation of the individual carrying hazardous material is a complex problem. The fusion of chemical sensor data with the data from optical cameras can be used for tracking of the suspect [36]. Accurate person localisation is achieved through the fusion of different sensor data in narrow corridor spaces. Once an individual carrying such material is detected and tracked, the sensors will share the information with other sensors and the control centre through the communication system.

#### 2.3.3. Concealed Weapon Sensors

The detection of concealed weapons is a challenge to conventional video and thermal cameras. Weapons are normally characterised by their shape, material and appearance. Millimetre (30–300 GHz) and sub-millimetre (0.3–3 THz) wave sensors are state-of-the-art detectors for concealed weapons. There are active and passive scanners based on the radar principle. Active scanners operate in the range of 350–800 GHz, direct waves on the subject and receive the reflected energy. Passive scanners receive radiation from humans and other objects and operate in the 30–350 GHz range. Compared to active scanners, passive scanners provide a low signal-to-noise ratio (SNR) and low-resolution images, which are more challenging to process. Figure 5 shows an example of using radio waves in public transport stations to perform safety checks.

#### 2.3.4. Health Sensors

Train stations are the entry and exit points of states and countries. The spread of an epidemic or pandemic, such as COVID-19, can be controlled or limited using health sensors. Such health sensors are deployed to classify whether individuals are healthy or not by measuring people’s vital signs, such as fever and heart rate. Infrared thermography (IRT) is currently used at airports for monitoring travellers’ fevers. Accuracy can be improved by using the same IRT sensor on screening based upon all vital signs [38]. The fusion of health sensor data with camera data can provide real-time tracking of a sick individual as shown in Figure 6.

#### 2.3.5. Acoustic and Vibration Sensors

Acoustic or sound sensors are the most commonly used non-conventional sensors. Acoustic and vibration sensors provide detection of humans and other moving objects, which is helpful for object tracking algorithms. Compared to imaging sensors, acoustic and vibration sensors do not require line of sight to provide omnidirectional coverage. These sensors can also trigger an event, such as turning on lights and imaging cameras for video surveillance. In addition, audio analytics helps video analytics detect target events such as screams, glass-breaking sounds, explosions, and gun fires.

### 2.4. Monitoring and Recording System

A monitoring system consists of a control room, monitor screens, and control devices. The number of monitoring screens depends on the number of cameras and their functionality. Some modern monitoring (IP-based) systems can be deployed on mobile devices such as smartphones and tablets, and even mobile devices provide limited functionality due to the device size. CCTV operations require big screens for optimum operation. There are two modern big screen monitoring technologies, liquid crystal display (LCD) and light-emitting diode (LED) displays, which have superseded the older cathode ray tube (CRT) displays and plasma displays. Table 4 compares these monitors. The following aspects should be kept in mind when choosing a display:Size: Size is usually specified as the length of the diagonal measurement of the monitor. The size is selected based on the number of simultaneous camera displays. As a rule, the operator to screen distance is three to five times the screen size.Resolution (Number of pixels): The number of pixels specifies the display resolution. The pixel number has a one-to-one correspondence to the resolution of the camera output image. For best performance, these should match.Aspect Ratio: Aspect ratio is defined as the ratio of the number of pixels in the horizontal to those in the vertical. For best results, the aspect ratio of the monitor should match that of the camera.

The recording system records digital images and auxiliary data, as well as their copies, on a digital recording device. Some copies are stored on other media platforms, such as Cloud platforms, for safety reasons. The recording system is an important part of the surveillance system since smart surveillance relies on recorded video data.

Traditionally, all the components of a CCTV, from sensors to recording and monitoring stations, are installed at the same location in response to the real-time request. However, the long-term installation and maintenance costs can be high. These costs can be reduced by utilising a distributed computing infrastructure. The users of the CCTV system usually set the regions of interest and the location of monitoring rooms, while the system designers typically set the location of the computing and recording systems. This computing infrastructure can be set up to reduce the costs in four configurations, as shown in Figure 7.

### 2.5. Video Analytics

Video analytics is defined as an automatic processing of video analysis aimed at assisting the surveillance operations. Some of the main challenges to video analytics for railway surveillance from a computer vision perspective can be subdivided into two big groups depending on whether objects are moving or static: (1) classification, detection, segmentation and decision making for static objects such as left bags or a person laying on the rails of the network; (2) classification, detection, segmentation, tracking and decision making for dynamic events such as crowd monitoring, tracking a terrorist that is running to escape, a thief or the suicide of a passenger.

Both groups of approaches have in common that they need to be able to provide reliable solutions under a wide range of conditions, both naturally occurring (e.g., lighting changes, weather condition changes) and cyber attacks, which include intentional interference during different stages of the imagery data transmission, processing and storage. Hence, there is a demand for a wide range of methods able to meet different requirements and provide trustworthy solutions. One way to provide a level of trustworthiness of machine learning approaches is to evaluate how the different above-mentioned factors, which we call uncertainties, impact the developed solutions. Are the developed solutions reliable, and if yes, under what conditions and circumstances? The answer can be found based on probabilities or other information measures representing different levels of trust in the developed solution.

In addition, there are several other challenges due to occlusions, environmental conditions, scalability—of the considered monitored area, and big heterogeneous sensor data. In terms of network rail stations, we have small, medium and big train stations for railway networks in big geographic areas.

Video analytics requires a combination of several computer vision techniques, which are introduced in Section 2.1.1. There is a wealth of video analytics methods as summarised on Figure 8 out of which machine learning methods are now an active area of research. Classical methodologies [40,41] for tracking the motion of the targets and detecting the change of the motion are implemented by firstly applying object detection techniques such as the Optical Flow [42,43,44,45] techniques to determine objects of interest within the scene and estimate their motion, and then using classification methods such as Gaussian Process Regression [46,47,48] to detect the change point of the behaviour.

In recent years, deep learning methods based on Convolutional Neural Networks (CNNs) [49,50,51,52], which are also widely used in building Artificial Intelligence (AI) have become the most popular computer vision technology. Different from the traditional machine learning methods, deep learning methods developed recently have provided more robust and accurate algorithms available for object detection and pattern recognition.

#### 2.5.1. Deep Learning Methods for Video Analytics

Convolutional Neural Networks (CNNs), Recurrent Neural Networks (RNNs), and Vision Transformer (ViT) are the three most popular deep learning methods used implementing video analytics models for video surveillance.

##### Convolutional Neural Networks

CNNs have become the most powerful tool for computer vision tasks due to their strong performance on image and video data. CNNs are biologically inspired, and they are built to learn invariant features with millions of neurons. Compared to a regular Neural Network, a Convolutional Neural Network layer consists of weights parameters in three dimensions: width, height, depth, which makes it possible to reduce the image features into a single vector by the end of an entire network’s architecture. In recent years, various deep learning models developed based on CNNs have been demonstrated with strong robustness performance on image datasets such as ImageNet [49], which makes it the most popular deep learning framework in the area of computer vision [52]. One of the most popular applications is face recognition [53,54,55]. A basic CNN structure is shown in Figure 9.

##### Recurrent Neural Networks

Recurrent Neural Networks (RNNs) [50,56,57,58,59] are another important class of networks for video surveillance. However, learning the image features in certain tasks requires a short-term memory. Hence, deep RNNs [50] are developed to address the problem where traditional neural networks are not able to learn the features of the input data from the previous data in a temporal sequence. RNNs are built based on loops of neural networks. Figure 10 shows the architecture of the loop structure inside RNNs. AI models built with RNNs allow them to understand the current features based on the previous data.

In recent years, RNNs have been widely applied for solving problems with sequential input data such as speech recognition, language modelling and image prediction [58,59]. However, in some cases, not only short-term memories are required, long-term dependencies are also needed. In response to this problem, Long Short-Term Memory networks (LSTMs) [60] are designed as a certain type of RNN, which is especially good at learning the long-term dependencies. Different from RNNs, the repeating module in LSTMs has several neural network layers, which are designed to be able to memorise the previous features in the cell state.

A convolutional LSTM is firstly used in building a spatio-temporal video autoencoder in future frame prediction [61]. The convolutional LSTM cells are used to build the temporal encoder to integrate changes over time. Deep neural networks are normally used as the method for learning the spatial representations. Recent works [62,63] utilise the advantages of LSTM networks with long-term dependencies to build Convolutional LSTM networks used to learn the temporal features of the input sequence video data, which significantly improves the performance of the proposed method of using an LSTM convolutional autoencoder for video anomaly detection.

##### Vision Transformer (ViT)

In addition to CNNs and RNNs, a novel deep learning structure called a *transformer*, which is based on a *self-attention* mechanism [64], has become a popular method used in the area of objective detection and behaviour recognition. A transformer is firstly developed for solving Natural Language Processing (NLP) tasks. Due to its strong representation capabilities and huge success with NLP tasks, researchers are expecting that transformers will be able to achieve huge success performing computer vision tasks as well [65]. Such improved transformers built for computer vision tasks are called Vision Transformers (ViT) [65]. A basic Transformer structure is shown in Figure 11.

When evaluated on computer vision benchmarks, AI models built based on transformers have achieved similar or even better performance than models built based on other types of classic networks such as CNNs and RNNs [65]. Transformers are gaining popularity among the computer vision community due to their strong performance and lack of the necessity for vision-specific inductive bias [66,67]. As a novel deep learning framework, ViT now has gradually become the most powerful method for image classification tasks.

### 2.6. Motion or Moving Object Detection

Motion or moving object detection (MOD) [68,69,70] is one of the earliest video analytics features made available in CCTV surveillance systems. The aim of MOD is to detect the event of a single object or multiple target objects of interest moving in the region of interest.

There are three traditional MOD approaches, namely frame difference [71,72], background subtraction [73,74] and optical flow. Frame difference relies on the temporal difference of pixel intensity in consecutive frames. The difference of pixels belonging to static objects is close to zero, and the moving objects are detected with high difference values. The moving objects in a static background are detected using a foreground detector. The optical flow approach measures pixel-level velocity vectors to detect the motion.

#### 2.6.1. Static Object Detection

Static object detection (SOD) [75,76,77] alarms a surveillance operator when an instance of an object of interest is detected in the scene captured by the cameras. The methods for static object detection are different from the moving object as there is no relative movement between the object of interest and the background. Most of the other video analytics processes such as tracking, face recognition, person re-identification, and many more rely on SOD as a pre-processing step.

Figure 12 shows an example pipeline of a two-step SOD. The object detection methods generally do not differentiate between objects of one type, e.g., the suitcase and the backpack are classified as one type, i.e., bag. In a single-stage detector, all these steps are performed by a multi-layer neural network. From a user point of view, all objects of interest are mentioned in the requirement document.

#### 2.6.2. Multiple Object Tracking

Multiple object tracking (MOT) [34,79,80] aims at identifying and tracking multiple objects of interest in a video without any prior knowledge of their appearance and disappearance from the scene. The video can be from the same or different cameras. MOT differs from SOD in the sense that each object will be given a unique label, which is maintained in all video frames by the MOT algorithm. The MOT algorithm will identify all objects, such as the example using Gaussian models shown in Figure 13.

Object detection, feature extraction, motion prediction and the association methodology are the key areas of research for static and multiple object tracking. State-of-the-art methods for object detection and feature extraction are deep learning methods [13].

State-of-the-art object detection methods are built based on deep learning models. Generally, there are two types of detectors: two-stage and single-stage detectors [82]. Two-stage models have a separate module to generate region proposals. In the first stage, the models aim to find an arbitrary number of object proposals. Then the objects will be classified and localised in the second stage. Single-stage detectors classify and localize semantic objects in a single shot using dense sampling. Several popular two-stage detectors are: R-CNN [83], Fast R-CNN [84] and R-FCN [85].

Faster-RCNN [86] is one of the advanced CNN-based deep learning objective detection methods with powerful performance. You Only Look Once (YOLO) [87], which is an even more powerful algorithm widely developed in recent years, is also used as the object detector ending up with a significant improvement for objective detection under different kinds of scenarios. These algorithms usually label the objects in the video by bounding box regressions, i.e., the tracking box shown in Figure 14. Then, other object tracking algorithms can be applied to the objects.

Figure 13 shows multiple detected objects, including vehicles passing through the traffic crosses and the pedestrians walking across the roads. In intelligent surveillance systems, Extended Kalman filters and Gaussian Mixture models are commonly used machine learning-based methods to track moving objects [88]. In classical methods, such as optical flow types and background subtraction, the minimum number of frames used to detect traffic flow disruptions is two, although the number of frames used can vary.

After detecting the objects using the object detection algorithms introduced above, then motion prediction is achieved by using Bayesian approaches such as Kalman filters, particle filters and probability hypothesis density filters. The association probability calculated using both the motion and feature information performs better than those which use one of them. Advance methods, such as the Multiple Hypothesis Tracking (MHT) framework, process the data of multiple images to associate labels. These methods are based on data from multiple time steps to give better results.

Figure 14 shows an example pipeline using the topic modelling method to detect the abnormal behaviours under the traffic scenario [81]. The minimum number of required frames using topic modelling to detect the target disruption in traffic flow is two. The disruption in traffic flow is detected when the discriminant function reaches the threshold. In rail network cases, SOD and MOT techniques can be used to detect target objects with abnormal behaviours that could cause potential harm to the safety of the rail system, such as a person leaving a suitcase on the platform. SOD and MOT are the basic techniques for Video Anomaly Detection, which will be introduced in the following section.

### 2.7. Video Face Recognition

Video face recognition (VFR) [53,54,55] deals with identifying an individual from the video data by comparing a given person’s video with a library of still face images. In rail networks, VFR technologies can be used to automatically raise the alarm when the sensors detect an individual from a specific watchlist. It can also be used to search for a specific person, through an image of the target from the recorded surveillance data. A basic VFR process is shown in Figure 15.

Before the identification stage, the captured video data need pre-processing. The pre-processing step contains face detection, face alignment, and face aggregation. Firstly, the face detector localises the faces in each frame. Then, the face alignment detects facial landmarks such as eyes, nose and other important features. Finally, the face aggregation represents the aligned faces in multiple frames by a comparatively smaller set of frames. As a result of aggregation, noisy and low-quality frames are rejected.

All VFR methods extract a facial feature set corresponding to a separate image before matching them with a library of feature sets. Each feature set in the library will be assigned an identification tag. The matching algorithm identifies the object through the differences between the feature sets extracted from the aggregated frames and stored in the library. Whether a feature library consists of single or multiple entries of the same person, the VFR performs a still-to-video verification process. Some VFR methods that can only identify objects already recorded in the library are called closed-set VFR. A model based on open-set VFR is preferred for such a security system as a railway station. For example, the open-set VFR can be implemented to identify people on a certain watchlist [90]. The VFR methods are categorised into two main streams, conventional and deep learning methods. Most of the modern VFR models that are already deployed at railway stations are built based on CNNs since they are more robust and accurate, and they are able to make decisions by themselves as compared to the conventional methods [9].

VFR helps in crime detection by tracking missing persons and estimating passenger traffic at any time in a rail network. Another application of VFR is that a facial recognition system could be used to replace the need for tickets on trains. Such systems have been tested in the UK [91]. By using two near-infrared lights helping a single camera determine the texture and orientation of each pixel, the smart surveillance system is able to successfully identify passengers’ tickets without the need for them to stop walking and show their tickets.

### 2.8. Person Re-Identification

Person re-identification (Re-ID) [92,93,94] aims at identifying a person across non-overlapping cameras and different modalities. It is also used for identifying an individual who passes through blind spots and re-appears in the coverage of the same camera. It serves various security purposes such as online tracking of individuals over multiple non-overlapping cameras and searching by a query from several cameras’ recorded video data. This query can be an image, a video or a text description of the person. Depending on the different components involved in the surveillance system, Re-ID can be categorised into closed-world and open-world settings [92]. The closed-world setting requires annotated searched video data and vice-versa for open-world settings. The open-world person Re-ID technology used in video surveillance systems is shown in Figure 16.

As shown in Figure 17, there are five main steps for building a person Re-ID system: raw data collection, bounding box generation, training data annotation, model training and pedestrian retrieval [92]. CCTV firstly detects images containing target individuals. Then, deep learning models provide feature extraction processes. The features extracted from the particular camera images are matched with those obtained from each library image through a matching algorithm based on the similarity. Potential targets and images can thus be filtered out through the matching stage.

The processings are the same when the given image is replaced with an individual’s video or textual description. The feature extraction is the key stage. In recent years, deep learning methods have been deeply developed with their great performance, but the current deep learning methods need further improvements for practical implementations [92].

### 2.9. Human Activity Recognition

Human activity recognition (HAR) involves classifying an individual’s activities or a small group of individuals. It is categorised into the following three types [11,95]:Gestures. These are elementary motions of the human body, such as stretching of arms or raising a leg.Action. These are the combinations of human gestures, for example, running and punching.Interactions. These involve interactions among multiple individuals and objects. These can be classified into human–object, human–human and human–object–human interactions. The human–object interaction deals with recognising the interaction of a human with an object, for instance, an individual abandoning an object or loitering. An individual interacting with another is human–human, such as a brawl. The interaction between two humans and an object leads to human–object–human activity, such as stealing a bag from another individual.

As shown in Figure 18, the output class is “Abandoned Object”. In addition to the classification, the alert also provides the object’s location by highlighting it on the screen (yellow box in the example). The state-of-the-art methods for HAR are deep learning-based methods that merge the above-mentioned two steps into a single unified framework [12]. The latest architecture of the deep learning methods is based on two streams, one for the spatial and the other for temporal processing [96]. The output of two streams is merged to give the final output. This two-stream architecture is also called the Spatio-Temporal Network (STN). The surveillance system user defines all the classes of gestures, actions, and interactions required to be automatically detected by the HAR algorithm.

### 2.10. Video Anomaly Detection

Video Anomaly Detection (VAD) aims to detect abnormal behaviour of an individual or a group of people through a camera or multi-camera video. An anomaly or an abnormal behaviour is difficult to define; normally it is a psychological term for actions that fall outside of the realm of what is considered normative in a particular society or culture. In this case of rail networks for safety purposes, it can be defined as an event or behaviour that could lead to a potential threat to public safety. VAD is an essential technology for safety surveillance systems.

In railway systems, VAD can help prevent trespassing and suicides at train stations. Different from HAR, VAD does not rely on a number of manually labelled training data to detect normal and abnormal behaviours. VAR has better performance than HAR in cases where the classifier does not have enough labelled data of abnormal behaviour occurrences. Individual unsupervised anomaly detection (IUAD) is useful for detecting a person attempting suicide [98] or a terrorist attempt [99] in the surveillance region, where it is assumed that the individual carries out the daily routine tasks abnormally with a suicide or terror motive in mind. A basic block diagram of the IUAD algorithm detecting an anomaly of a person falling on the floor is given in Figure 19.

Depending on how the algorithms are built based on training data, there are three main categories of anomaly detection methods [100]: unsupervised methods, weakly supervised methods and supervised methods. Some of the anomaly detection tasks can be performed using supervised or semi-supervised anomaly detection methods. The supervised anomaly detection model is trained on labelled data of normal and abnormal behaviours. The semi-supervised anomaly detection model is trained on labelled data of normal behaviour. The unsupervised anomaly detection model trains on unlabelled data, such as real-time surveillance videos, which is a more practical and dynamic approach. Since the abnormal detection is supposed to be able to recognise the anomaly events with low happening probability, unsupervised algorithms can be considered as novelty detection with self-learning ability, which aims to cluster the outliers from the normal points. These methods provide a better generalisation ability under a single scenario with a fixed surveillance camera of the unknown anomalies.

Some works categorise VAD methods, specifically with respect to their final objectives instead of network architecture or learning strategy, into two categories, namely using reconstruction error or reconstruction-based methods and predicting the future frames or prediction-based methods [101]. Some of the recent deep learning methods [102,103,104] focused on implementing the video anomaly detection by reconstructing the predicted frames based on the features extracted from the previous frames through an autoencoder model. Abnormal events are located by the calculated reconstruction errors. When the reconstruction errors are over the threshold, the event happening in the current frame is considered an abnormal event.

Two methods built upon the autoencoders are proposed to generate a model for computing the irregularity error [102]. An autoencoder [105,106,107] is a feedforward neural network built for denoting a process of encoding the input into a more compact representation and then reconstructing the input with the learned representation as to the output. The irregularity error is implemented through two stages: first is to train a fully connected autoencoder on the spatio-temporal local features. Secondly, build a fully convolutional feedforward autoencoder to learn the classifier. To improve the performance of the Autoencoder, LSTM Convolution Neural Networks are leveraged for memorising the motion features of the past frames [103]. The proposed Convolutional LSTM Autoencoder framework has better performance with respect to encoding the change of the motions of the objectives for normal events.

As the deep Autoencoder has been widely applied for anomaly detection, a new autoencoder model with a memory module called Memory-augmented Autoencoder (MemAE) is implemented [104]. The traditional Autoencoder is improved by memorising the model parameters trained from normal samples at the training stage. Then, the reconstruction error is obtained while updating the rest of the autoencoder model at the test stage. MemAE has been proved to have better generalisation ability and higher effectiveness than the normal autoencoder algorithms. A novel deep Predictive Coding Network (AnoPCN) consisting of two modules: a Predictive Coding Module (PCM) and an Error Refinement Module (ERM) is introduced in [108]. Similar to LSTM, the feedback connections of PCM carry frame prediction, and the feedforward connections are built for computing the prediction error. AnoPCN has been verified with state-of-the-art performance on experimental data sets.

Instead of using an Autoencoder to compute the reconstruction error, there are other deep learning frameworks built to predict future frames. A method was developed that uses Generative Adversarial Networks (GANs) to train a generator, and alternatively, a discriminator is used to achieve the future prediction [100].

### 2.11. Trajectory Analysis

The MOT localises multiple objects across time. The path of each object is called a trajectory. Analysis of trajectory, also called Trajectory Analysis (TA) [109,110], provides valuable intelligence in the domain of video surveillance. Short-term trajectory analysis detects abnormal events such as walking and fighting. The long-term analysis can detect events such as loitering and interactions. The analysis can also be extended to analyse group activities by analysing multiple relative motions.

Combining TA with prior regional information provides detection of important events. For example, marking out-of-bounds areas in the TA alarms for intrusion detection and integrating the station layout with TA can detect pedestrian falls on the tracks. TA is also used for generating a summary or synopsis of a video. A video synopsis or summarisation aims to shorten a video by selecting critical frames. Fast-forwarding is another way of shortening a long video, but this method may miss essential frames.

### 2.12. Crowd Analysis

Crowd analysis [111,112] aims at understanding the behaviour of a crowd. A crowd is defined as a group of people behaving coherently. In a train station, travellers often move in groups, which makes it challenging to perform high-level processing such as MOT and anomaly detection. In such scenarios, instead of tracking and analysing each individual, the tracking and analysis of the crowd are feasible. Crowd analysis also aids in crowd management under unforeseen circumstances, such as emergency evacuation. The following types of crowd analysis are essential in video surveillance [16,113]:

Crowd Count (CC) aims at estimating the number of people from a video of a region of interest. CC enhances situational awareness in crowded areas. An increasing number of people in an area requires extra attention for management and safety purposes in a railway station. Moreover, CC can be used for gathering statistical data of visitors in different areas. The conventional methods for crowd counting rely on detection and regression techniques. Detection methods perform worse when there is a dense crowd and high background clutter. Counting by regression extracts low-level features and maps them onto a crowd count without considering spatial information. A block diagram of a typical density-based CC method is shown in Figure 20.

Figure 20 shows an example process of how CC algorithms deal with the input videos consisting of small groups of people. A more intuitive output of the CC algorithm is the congestion level. The crowd count is mapped on a scale ranging from sparse to dense crowd level. An alarm will rise when the crowd density rises beyond a critical threshold, which can cause dangers such as a stampede. One state-of-the-art method for crowd counting in videos is called the deep Spatio-Temporal Network (STN) [17].

Crowd Behaviour Recognition (CBR) identifies specific actions and events of a large gathering or crowd. Identification of different crowd behaviours helps in preventing a difficult situation. It can also be used to manage the flow of pedestrians. For example, bottlenecks and blocking of pedestrian flow are recognised using CBR [114]. A typical block diagram of CBR is similar to that of HAR, shown in Figure 18, where instead of extracting features of individual objects, the crowd features are extracted. Crowd Anomaly Detection (CAD) deals with the detection of abnormal behaviour in a crowd [115]. Similar to VAD, CAD helps automatically detect suspicious activities in crowded scenes and helps prevent a security-related incident from happening.

## 3. The Impact of Uncertainties in Computer Vision Technologies for Surveillance Systems

In addition to the challenges that exist in the current methods of CCTV surveillance, including the accuracy and efficiency of the results, there is another research area of the uncertainty in deep learning methods, which is the potential challenge of the security of surveillance systems. Investigating whether the results output by AI models are reliable is essential to security-related systems. For surveillance systems in rail networks, a practical and comprehensive AI model with the ability to think can provide trustworthy results for detecting the abnormal behaviour of people in public scenarios.

Most deep learning models introduced in Section 2.1.1 are considered as deterministic functions, which are barely used as solutions for possessing uncertainty information compared to probabilistic models. Probabilistic Graphical Models (PGM) use diagrammatic representations to describe random variables and relationships among them [116], which provide us with a practical tool for investigating the uncertainties inside the neural networks. Different from deterministic deep learning methods relying on estimations of the weights parameters and predictions, probabilistic models instead infer the distributions of the weights to reserve the uncertainties. A Gaussian Process [46,48] is one of the classic methods that define probability distributions over functions. This machine learning probabilistic method provides confidence bounds for data analysis and decision making—reliable information that, for instance, allows a biologist to analyse data or an autonomous car to decide whether to stop or not.

### 3.1. Uncertainty in Deep Learning

Deep learning models are designed to be able to return prediction results with high confidence when given data under the same category as the training data. However, the data fed to deep learning models can often be different from training data. These kinds of data are called out of distribution test data [117]. Under such circumstances, an ideal model would not only be able to return a prediction but also be capable of returning an analysis with additional information on whether the result lies outside of the data distribution. There are a number of factors that could lead to inaccurate solutions:(1)Noisy data/out of distribution data.(2)Uncertainty on the deep network parameters that are chosen during the training stage. This is also known as uncertainty in model parameters.(3)Model structure difference depending on the chosen algorithm for building the model (also called structure uncertainty). This uncertainty can be reflected in the following ways: an absence of theory and causal models and computational costs.

#### 3.1.1. Types of Uncertainty

Generally, there are two main types of uncertainty: aleatoric uncertainty and epistemic uncertainty [104,117,118]. Aleatoric uncertainty, also known as data uncertainty, refers to the irreducible uncertainty caused by the inconsistency between the data used in training the model and the out of distribution data mentioned above. Compared to an inherent property of the data distribution, aleatoric uncertainty is more like a property of the model, which makes it irreducible. On the other hand, the uncertainty caused by model structure and inadequate knowledge is called epistemic uncertainty [117], which can have a great impact on various uncertainties related to the choice of model structure.

Efficient estimation of these two types of uncertainties can be achieved within the framework of Bayesian neural networks [119]. One way to accomplish this is in the last layer of the Bayesian neural network and by providing the mean and variance, e.g. over the image classification result. The method of Kendall and Gal [119] estimates the aleatoric and epistemic uncertainties by constructing a Bayesian neural network with the last layer before activation consisting of mean and variance of logits. In this method $f^{\omega }\left(x^{☆}\right)=\left(\mu ,\sigma^{2}\right)$ represents the last 2*K*-dimensional pre-activated linear output of the neural network, where μ and $\sigma^{2}$ are the mean and the variance of *K* nodes. For the realised weights ${\left\{\hat{\omega_{t}}\right\}}_{t=1}^{T}$, with new input $x^{*}$, predicted mean and variance denoted by hat, with corresponding outputs $f^{\hat{\omega_{t}}}\left(x^{*}\right)=\left(\hat{\mu_{t}},\hat{\sigma_{t}^{2}}\right)$ [120], an estimator calculates the predicted uncertainty as a sum of the two types of uncertainty
(1)PredictedUncertainty=AleatoricUncertainty+EpistemicUncertainty.

There are different ways to express the aleatoric and epistemic uncertainty terms from (Equation 1) and these aspects are currently a subject of intensive research.

The next subsection discusses common adversarial attacks that are intentionally introduced to disturb deep learning methods.

#### 3.1.2. Adversarial Attacks

In addition to the uncertainties introduced above, another type of uncertainty can be added intentionally into models, which could disrupt significantly security systems such as surveillance systems. These uncertainties can be caused by changes made to the video or image data, which are often called adversarial attacks [121,122,123,124]. These intentional disturbances could have a great impact on how much people trust the results output by the models. As shown in Figure 21, even adding a little change to the original image, the result can be completely different. Many machine learning methods, including state-of-the-art neural networks, are vulnerable to adversarial instances [118].

##### Adversarial Samples

An example showing that a machine learning model makes an incorrect prediction as a result of tiny, deliberate feature changes is shown on Figure 21. These changes are created by diverse technologies that are especially effective for machine learning models, while they are not comprehensible to human eyes. Adversarial attacks are normally formed by intentionally added small perturbations, which can cause an incorrect result with high confidence [125]. These perturbations are commonly too small to be perceptible for deep learning models. Moreover, the same image perturbations can fool different kinds of deep learning network classifiers [126].

Adversarial attacks can be generated intentionally in surveillance systems in rail networks. This technology is a huge potential hazard to security monitoring systems. For instance, adversarial attacks can be used to turn the abnormal behaviours detected by the camera sensors into normal behaviours. It also can be used to fool the facial recognition systems, turning a dangerous criminal into another totally different person, or even worse, breaking into surveillance system centres to achieve full control of the monitoring systems.

Generally, there are two stages, the training stage and testing stage, where the adversarial attacks can disturb the final results of the deep learning models. White-box attacks and black-box attacks are two types of adversarial attacks in the testing stage [128]. White-box attacks refer to the circumstance where the adversaries have the full acknowledgement of the target models, including its structure, data distribution, and model parameters. However, black-box adversarial attacks can create adversarial examples without access to the gradient and parameters of the underlying deep neural networks. Different defence strategies are more efficient when facing different types of attacks [129].

#### 3.1.3. Impact of COVID-19

The CCTV systems are planned, designed and deployed to keep the objectives and the assessed risks in view simultaneously. Sometimes, unforeseen events occur that are not expected to be handled by the system. The spread of COVID-19 and its impact on the world is a good example of such a rare event. The railway network has been one of the worst-hit industries, as the way we travel, visit or commute through the station was radically changed. During the pandemic, a range of options by which CCTV and related technologies could assist has been developed. Examples from around the world are summarised in this section.

Early in the pandemic, airports began routine walk-through temperature checks for passengers [130], and this is a readily transferable approach for a CCTV system with thermal camera capability (however, an estimated 46% of positive cases would not be detected). If audio recording is available, methods developed by researchers such as at the Carnegie Mellon University [131] or Cambridge University [132] could be deployed based on audio samples of COVID-19 positive and negative patients. The research aimed to detect a COVID-19 patient based on cough, breathing, and voice. The reported accuracy of one such algorithm is above 90% [133]. Face masks and social distancing have become mandatory measurements in many countries to prevent the spread of the virus. Some aspects of video analytics, such as VFR, are affected by the introduction of face masks. The accuracy of the state-of-the-art DNN-based VFR algorithms relies on how well the training data represents real data. These algorithms are trained on faces without masks and may give inaccurate results for masked faces. To erase these uncertainties caused by the inconsistency between training and test data, AI models need to be equipped with advanced deep learning algorithms such as Bayesian Deep Learning methods. Few public datasets of faces with masks are available [134], which can be used to re-train the AI models. A proposed VFR algorithm has improved the accuracy of face recognition with and without masks to 95% [134].

The CCTV system could also automatically detect people with and without masks using video analytics. A DNN-based algorithm, named RetinaMask, can provide greater than 90% accuracy in detecting and localising the face and the mask [135]. Some examples of the detection results are shown in Figure 22. The green and red boxes indicate a face with and without a mask, respectively. The algorithm was shown to perform accurately in confusing situations, e.g., faces covered with hands, hair or even tissues are detected as faces without a mask (red box). Audio analytics has also been used to detect if an individual is speaking through a face mask or not [136].

Other algorithms include crowd analysis to detect whether social distancing is under observation and raise the alarm for areas when it is too crowded for distancing to be effective. However, these are highly dynamic situations. In most cases, observation through surveillance is more useful for understanding trends in behaviour at the population or location level rather than supporting an immediate response. Any response would also need careful consideration of the legal basis of any intervention.

### 3.2. Uncertainty Quantification with Bayesian Deep Learning Methods

Probabilistic models, also known as Bayesian models, are the main methods for formulating the uncertainty [137,138]. The core idea of Bayesian modelling is, instead of calculating the certain values of weights parameters in neural networks, inferring posterior distribution of weights parameters by Bayesian probability theory. Thus, the certain prediction result output by deep learning models will be replaced by a result with a confidence interval. Combining with a particular rule such as an uncertainty score, a deep learning model with Bayesian methods will be able to tell whether the output result is reliable [139,140,141,142,143,144].

The main challenge of machine learning models such as the Gaussian Process is that they do not scale to big data very well, which makes them not applicable to deep learning models with millions of weights parameters. In order to investigate the uncertainties in computer vision tasks, which normally would have a significantly big image or video data, Bayesian Deep Learning (BDL) methods [116,145,146] are the solutions.

Different from CNNs, Bayesian Neural Networks (BNNs) [138,147] aim to learn the proposed posterior distribution over the weight parameters. The history of BNNs can be traced back to the 1990s [137,138]. These works extended the proposed models to small data sets, providing a robustness to over-fitting problems and uncertainty estimates. Over the decades, a lot of works have been focused on improving the scalability and explainable uncertainties of the existing deep neural networks [47,48,147,148,149].

In Bayesian Neural Networks (BNNs), instead of calculating the certain value of weight parameters ω in neural networks, the goal of the optimising process is to build the prior distributions p(ω) by following the Bayesian theory and then using this distribution to estimate which parameters are most likely to have created the data. To capture epistemic uncertainty in a neural network, [119] assumes a prior distribution over the weights, e.g., p(ω) can be chosen to be a Gaussian distribution. The *posterior* distribution over the space of parameters by the Bayesian theorem can be denoted as [118,119]
(2)$$p\left(\omega |X,Y\right)=\frac{p(Y|X,\omega )p(\omega )}{p(Y|X)}$$
where for classification prediction purposes $X=\{x_{1},\ldots x_{N}\}$ is the training input data and their corresponding classes $Y=\{y_{1},\ldots y_{N}\}$, p(Y|X,ω) is the *likelihood* function. p(ω) is the known *prior probability*. The aim is using (Equation 2) to optimise ω to produce the desired classification output. However, in practice the true posterior distribution p(ω|X,Y) given by Equation (Equation 2) is hard to evaluate analytically in most real cases. The solution is to define an approximating variational distribution $q_{\theta }\left(\omega \right)$ to be as close as possible to the original posterior distribution.

Variational inference (VI) is the basic approximate inference method for approximating the posterior distribution. However, the biggest challenge of VI is that it is difficult to apply to complex deep learning models aiming to solve real-world problems. Based on VI, more research has been focused on applying various advanced approximate inference methods to deep learning models to solve more complex tasks. Some popular researches are Bayes by Backprop [137,150,151,152,153], Monte-Carlo Dropout (MC Dropout) [151,154], Variational Autoencoders (VAE) [105,155], Natural-Parameter Networks (NPN) [156], Bayesian GAN [157,158], and Subnetwork Inference [159].

To avoid incorrect predictions when performing deep learning algorithms, it is important to take uncertainty into account. Bayesian deep learning methods have been applied to solve various computer vision tasks, including but not limited to image/video retrieval [139,160], depth estimation [140,141], object detection [103,161,162], semantic segmentation and scene understanding [142,163,164,165], optical flow estimation and motion prediction [143,166,167], human pose estimation and pedestrian localisation [144,168,169], person re-identification and face recognition [170], camera re-localsation [171], and avoiding adversarial attacks [172,173].

The uncertainty measurement abilities of Bayesian methods have been demonstrated to be useful for accurate object detection [103]. Since large-scale object detection datasets have ambiguities when labelling the bounding boxes, a novel bounding box regression loss function for learning bounding box transformation and localisation variance is proposed to improve the accuracy of localising the objects.

Bayesian Deep Learning [116] offers a promising framework to handle the two types of uncertainties in a general way. Since deep learning networks are mostly over-parameterised, it is a great challenge to handle the uncertainties efficiently in such a large parameter network. By placing a prior over hidden units, which composes the neural networks, Bayesian deep learning can reduce the overfitting problem with advanced regularisation methods, especially when we have insufficient data [154].

## 4. About the Next Generation of CCTV Surveillance Systems for Railway Stations

### 4.1. Data Centre

Often, advanced computing and recording systems are located at the periphery of train stations or at remote control centres [174]. The control centres communicate, store and process the big data from sensors. These centres also provide simultaneous storage and software services to multiple CCTV systems. The multiple sensor data are processed, fused, analysed and communicated to the CCTV operator through the monitoring system with appropriate warnings and decision cues in real-time. The concept is shown in Figure 23.

Future CCTV surveillance will have additional capacities, including augmented and virtual reality systems, advanced cloud computing and autonomous recognition and warning technologies. Figure 23 shows that such a CCTV system could be installed in two geographically distant stations communicating with a data centre for data processing and storage. This architecture enables economical use of the resources with the concept of (common) data centres, which are expensive to deploy and maintain. With the fast development of Cloud Computing and 5G networks [175,176] and technologies combined with AI, the function and performance of surveillance systems will be significantly enhanced. Under a deployed 5G network, the data recorded and developed by live sensors could be transformed fast and wirelessly and then stored in a Cloud Centre. A data centre will be an essential part of future surveillance systems.

### 4.2. Data Analytics

Another important aspect of future CCTV systems is the need of various and efficient methods for analysing the different types of sensor data both in real time and off-line. The data analytics are divided into four intelligence levels, as shown in Figure 24. The intelligence level increases from the bottom to the top, i.e., detection to decision making. The detection is provided by both conventional and non-conventional sensors. The higher-level intelligence is achieved through multisensor data fusion. The video data are important from the presentation point of view as the fused data are presented to the operator on top of the video data. The recent research is focused on autonomous and semi-autonomous detection, tracking, behaviour understanding and activity recognition [33,177,178]. The state-of-the-art data analytics are even smart enough to understand some types of scenes and record the time duration. The future decades will see progress in scene understanding and decision-making. The CCTV systems should be installed while keeping in mind the frequent software and hardware updates of the data analytics system.

The data presentation and control in the monitoring systems is another domain that will see a lot of enhancements in the future. With the increase in the number and modalities of the sensors, the monitoring stations would be upgraded for ease of the security operations. In current systems, the data of all the sensors are presented either on separate screens or in separate windows of the same screen. The operator manually selects different cameras to obtain situational awareness. For this, the operator remembers the building layout and the location of each camera to relate the video to the location within the region of interest. This task requires mental capacity, training and experience.

Future systems may blend real-world data feeds with 3D virtual models of an environment to provide the 3D situational awareness of the sensor data. Such systems are called augmented virtuality approaches [179]. Localisation and tracking of data can be matched between the real and virtual worlds.

The integration of a 3D virtual model of a real environment with sensor feeds from that real environment can draw inspiration from the ideas in a geographic information system (GIS). The sensors are geolocated and produce georeferenced data. On detection of an important event, a geolocated movement sensor, for example, could signal the main system to activate the corresponding CCTV camera for video coverage and recording and also engage any available operator’s attention. Another option would be for an operator to select an area of an environment and for the system to automatically direct all relevant sensors on that area, e.g., PTZ cameras. In addition, event prediction in blind zones could be performed, as illustrated in Figure 25.

Rail networks, especially railway stations, are an important social and economic pillar. While it is important to take state-of-the-art security measures when installing the latest CCTV systems, it is important to keep track of the financial implications. The best solution requires reasonable emerging technologies for the future systems in mind. In addition, cybersecurity measures need to be considered when implementing a surveillance network.

## 5. Ethics and General Data Protection Regulatory

Computer and information technologies can bring advantages to not only railway networks but also other computer smart cities and public transportation systems, while they will also raise ethical concerns about computer and information security and privacy [181]. Due to the fast development of new networked technologies, such as facial recognition (FR), some research aims to find and raise public awareness of ethical issues lying in urban big data analytics and public transportation systems [182,183,184,185]. Some researchers have been working on providing ethical frameworks for big data and smart cities focusing on contemporary ethical and non-ethical issues in big data analytics applications in smart cities and public transportation systems [183]. There are four emerging issues considered to exist commonly in all smart cities: privacy and surveillance, data integrity, social equity and the ageing population and data collection and analytics in public transportation [183].

The work of video analytics in rail network surveillance relies on the video data collected through CCTV systems. As video surveillance is enhanced due to the advancements in technology, it can produce a vast amount of personalised raw and analysed data. One of the main ethical concerns in the domain of CCTV is privacy. For example, video data used to identify a person using facial and gait features can cause a leak of personal information. Additionally, video analytics could identify an individual’s interactions with other individuals and objects. It helps in automatic monitoring on the one hand, but individual privacy is compromised on the other hand. Personal privacy is an important component of human rights, but maintaining safety and security in justifying an interference is subject to evolving interpretation. The privacy breach is not the only ethical consideration. The sense of being watched by a CCTV system could also constrain individual decisions and behaviour. Moreover, individuals may also be worried and act nervously to avoid being falsely detected as suspicious individuals.

In 2018, many countries enacted a law to protect individual privacy. The European Union General Data Protection Regulation (GDPR) took effect based on the six basic principles [186], as given in Figure 26. The UK implemented GDPR through the Data Protection Act (DPA) 2018, subsequently amending this to UK-GDPR. Under such rules, video data are considered personal data when it can be used to identify the person.

Smart surveillance is an area of evolving interpretation, particularly for automated facial recognition. In the UK, the Information Commissioner gave an opinion about the use of live facial recognition in 2019 [187], and in 2020, the use of this by the South Wales Police was ruled unlawful [188]. Another aspect of ethical concern is ensuring the use is for identifying specific people of concern rather than conducting mass surveillance and demonstrating that the software used does not have a racial or gender bias in its identification processes. Knowledge of the training datasets used in the production of the facial identification software would be crucial in ensuring freedom from these biases, indicating the degree of diligence required of end-users before the legal operation of such systems.

## 6. Conclusions

This paper provides a survey of the latest trends in CCTV systems and presents a vision for the future of such systems. The ever-evolving nature of security hazards and threats can be countered through analysis and fusion of multi-modal sensor data. The CCTV system produces big data, which can overwhelm the operator if not presented efficiently. Georeferenced monitoring systems with an augmented virtuality display can ease the burden and improve the response of the operator. The document gives a detailed account of the video analytics aspects that aid in the security and management of the station. The data analytics of other sensors, such as audio and thermal sensors, can further enhance the capabilities of the CCTV system. The processing and storage of the big sensor data cannot be met through a centralised computing architecture. The alternative architectures, namely cloud, cloud–fog and cloud–fog–edge, can be implemented for cost-effectiveness. Prior to the operation of automated systems, consideration must be given to ensuring outcomes are free from racial or gender bias imparted by poorly selected training datasets and that personal data are handled according to legal requirements of data privacy.

## Figures and Tables

**Figure 1 sensors-22-04324-f001:**
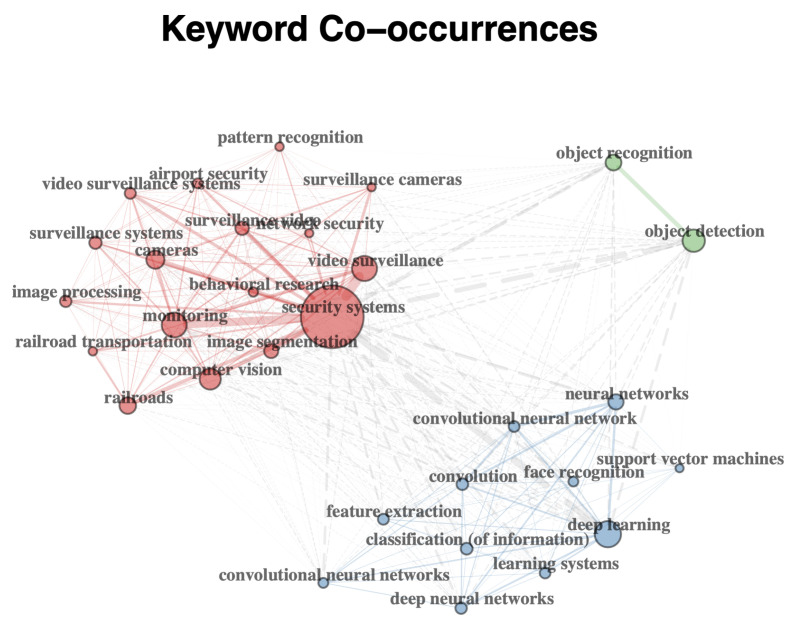
Keyword co-occurrence network of the published papers under the topic of video surveillance analytics published since 2010.

**Figure 2 sensors-22-04324-f002:**
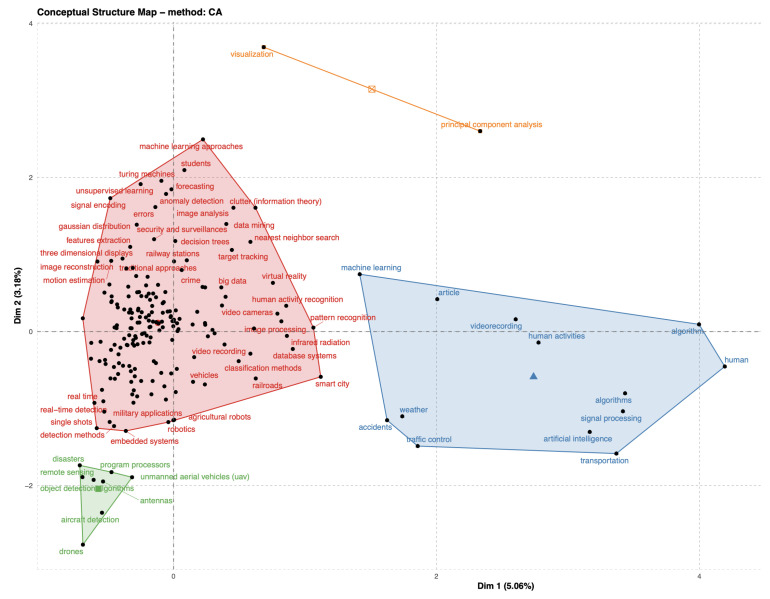
Conceptual structure map using the Correspondence Analysis (CA) method. The clusters represent how ideas are connected; the closer they are, the stronger the association.

**Figure 3 sensors-22-04324-f003:**
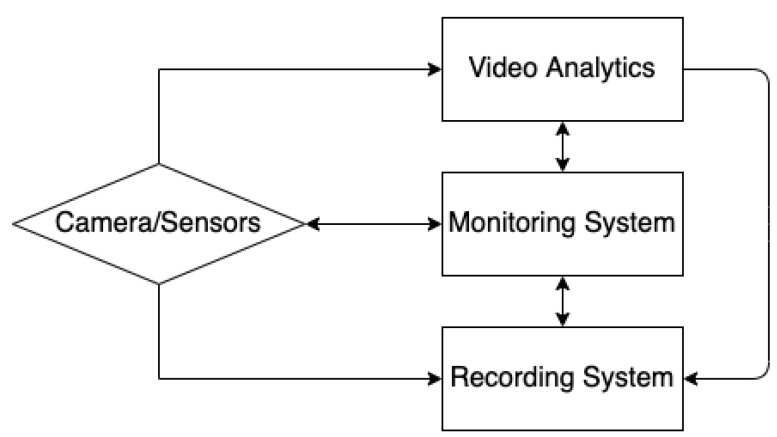
CCTV modules. In railway surveillance, cameras are the main sensors, which are responsible for collecting video data. Different types of sensors (introduced in Section 2.3), communication systems and other computing facilities (introduced in Section 2.4) make up the monitoring and recording systems. The video analytics module receives the data passed from monitoring and recording systems and then makes decisions using computer vision technologies.

**Figure 4 sensors-22-04324-f004:**
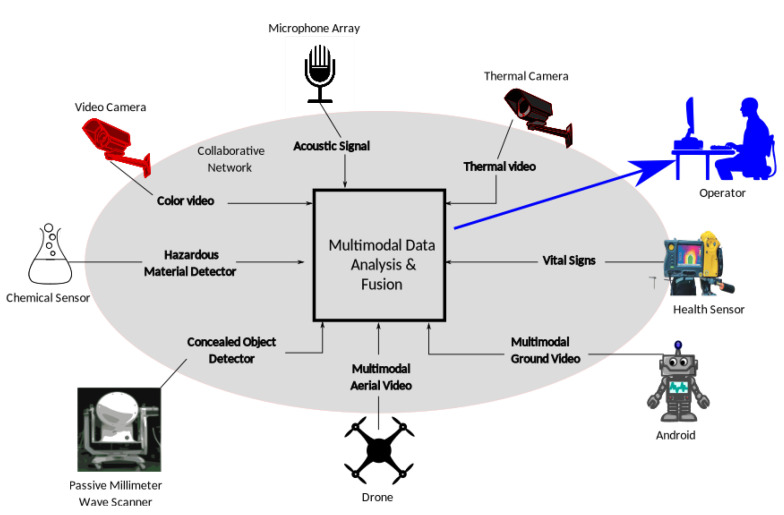
An example of a sensor system composed of conventional sensors, such as electro-optical and thermal imaging sensors, and non-conventional sensors, which enhance the capability in other frequency domains. Acoustic sensors provide omnidirectional detection and tracking based upon trilateration from temporal differences in the sound waves. The health sensor provides vital signs of the individuals for detecting sick people passing through a station for further screening. A chemical sensor detects specific chemicals for the identification of possible explosive devices. The wave scanner detects concealed objects, including weapons.

**Figure 5 sensors-22-04324-f005:**
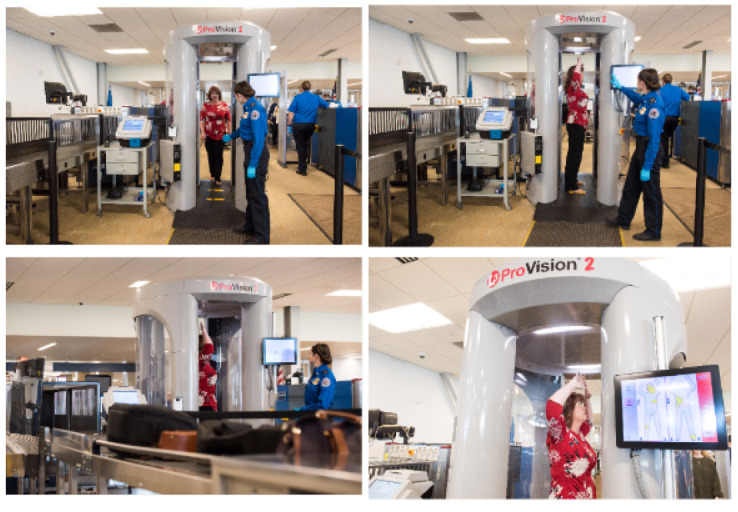
Using radio waves in the millimeter spectrum to safely penetrate clothing and reflect off body-worn concealed threats. Images were taken from the “L-3 Airport Scanner” by the Pacific Northwest National Laboratory [37].

**Figure 6 sensors-22-04324-f006:**
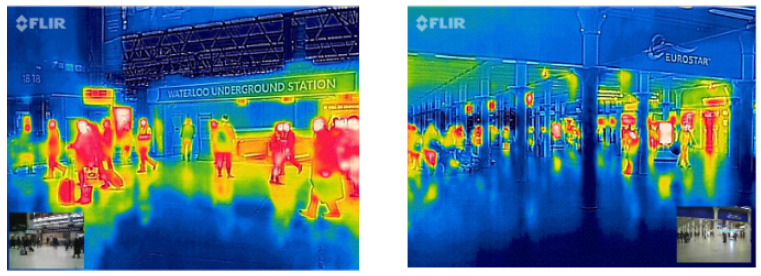
Infrared thermography used in a train station. Images taken from “Waterloo station” by branestawm2002 [39].

**Figure 7 sensors-22-04324-f007:**
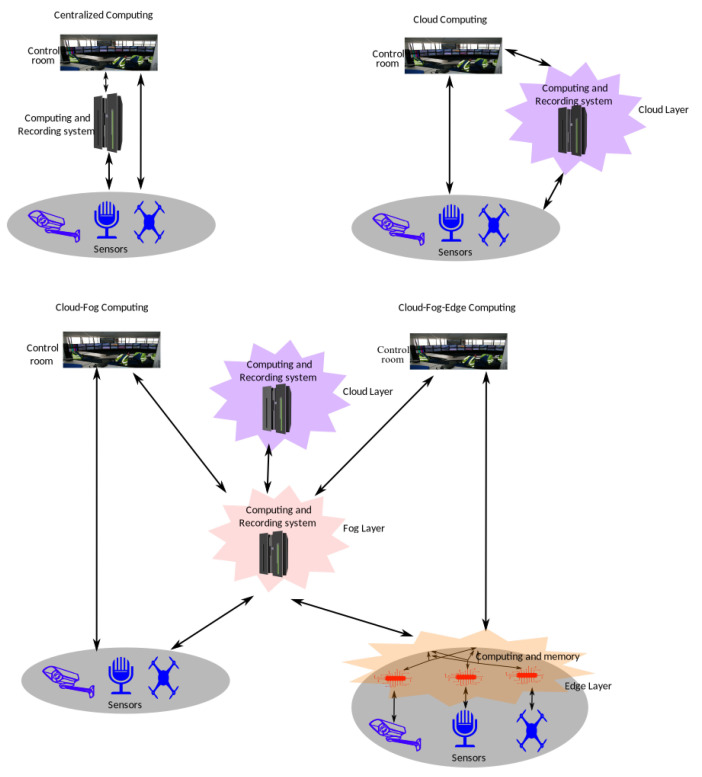
Computing architectures.

**Figure 8 sensors-22-04324-f008:**
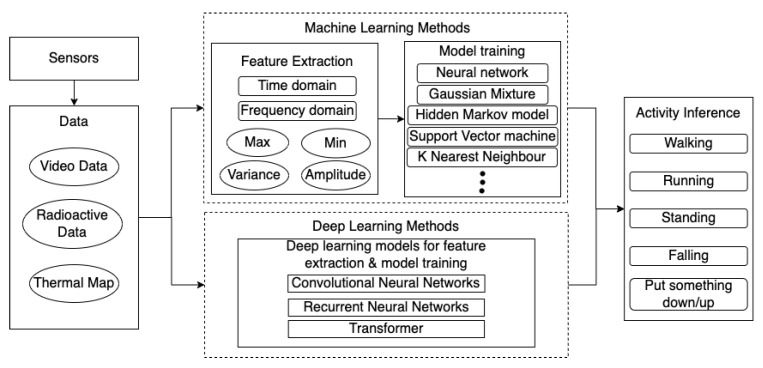
Block diagram of sensor-based computer vision methods for object detection and behaviour analysis tasks. The data collected by sensors will be transformed as input data to the train models. Models built based on various structures and methods will handle the process of extracting the features from the training data set. Then, well-trained models can be used as a detector or classifier to recognise the activities of the detected objects in surveillance systems.

**Figure 9 sensors-22-04324-f009:**
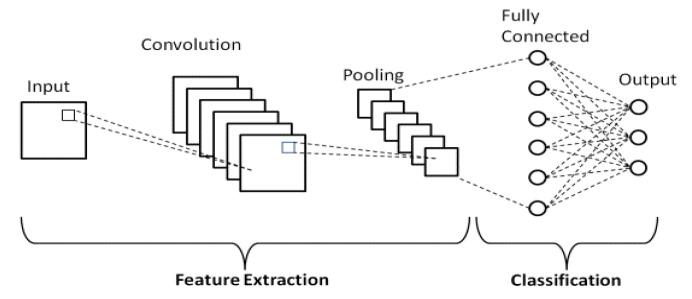
A basic CNN structure used for classification tasks. Generally, there are three types of layers used to form a full CNNs architecture: Convolutional layer, Pooling layer and Fully-connected (FC) layer. A convolution layer is a linear transformation that preserves spatial information in the input image. It will compute the output of neurons that are connected to local regions in the input, each computing a dot product between their weights and a small region they are connected to in the input volume. Pooling layers take the output of a convolution layer and reduce its dimensionality under certain rules. FC layers are used to connect the neurons between two different layers. It consists of the weights and biases along with the neurons. The FC layers are usually deployed before the output layer to work on the last few layers of a CNN architecture.

**Figure 10 sensors-22-04324-f010:**
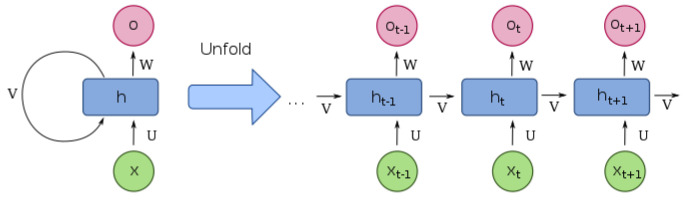
A diagram for a one-unit RNN. From bottom to top: input state, hidden state, output state. U,V,W are the weights of the networks. The compressed diagram is on the left, and the unfolded version is on the right.

**Figure 11 sensors-22-04324-f011:**
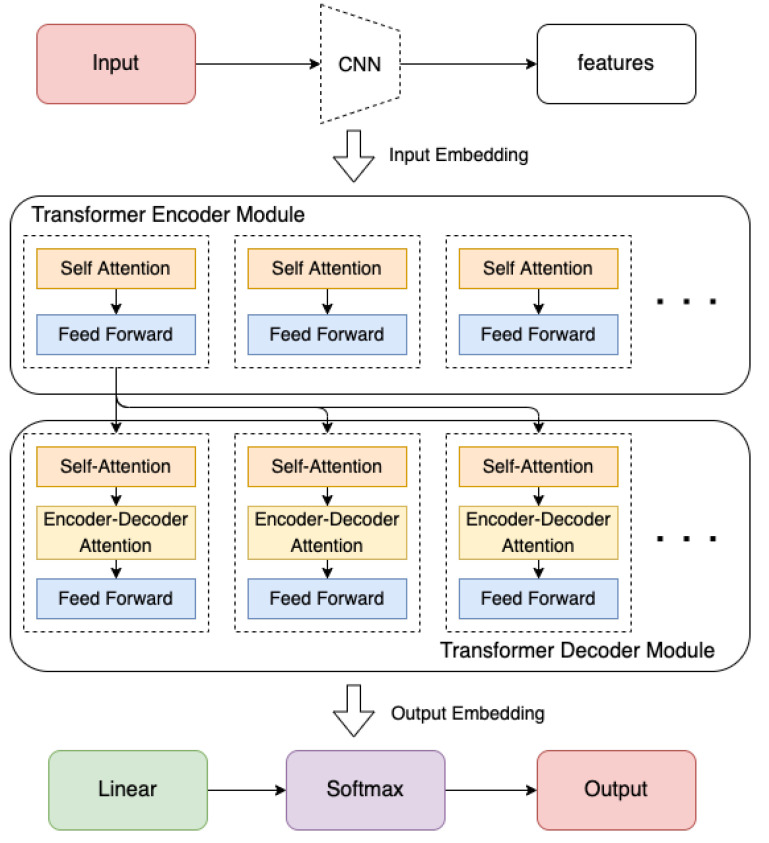
Pipeline of a Transformer [64]. A transformer is made up of an encoder module and a decoder module with multiple identical encoders and decoders. Each encoder and decoder contains a self-attention layer and a feedforward neural network. Each decoder has extra an encoder–decoder attention layer.

**Figure 12 sensors-22-04324-f012:**
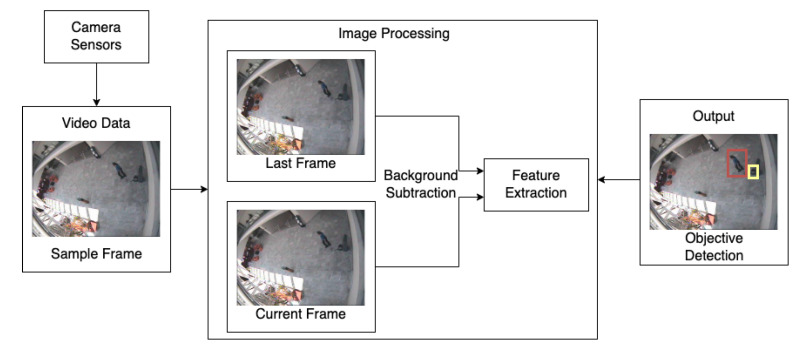
A two-step SOD pipeline using Background Subtraction. The images are from the CAVIAR dataset [78]. By comparing two different frames in the video data provided by the same camera sensor, the input image is divided into multiple region proposals based on the objects of interest. Each region is processed separately, and the object features in each region are extracted. The background subtraction detects all moving objects from the background. Then, by continuing to compare other frames in the video, the static object in the frame apart from the moving objects, such as people and trains, will be separated from other objects. Therefore, the static objective—in this case, the luggage—will be successfully detected.

**Figure 13 sensors-22-04324-f013:**
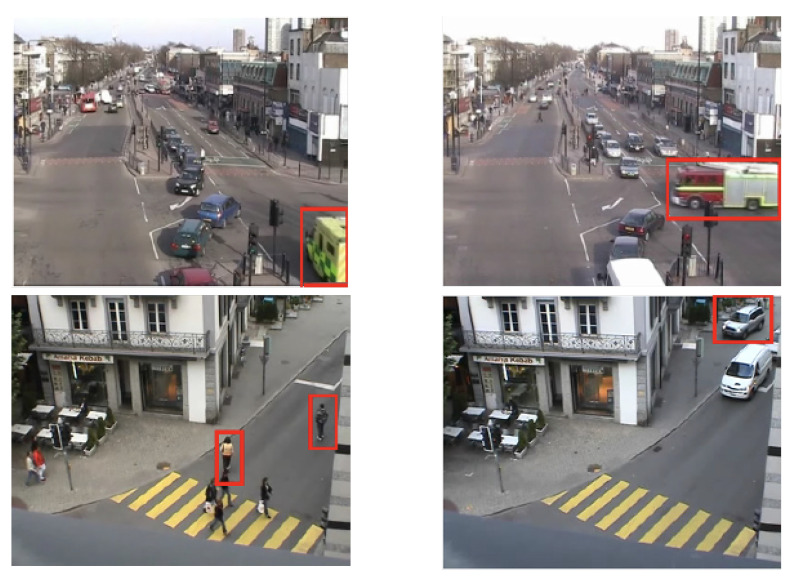
Multiple object detection examples [81]. The targets are detected and marked in red boxes.

**Figure 14 sensors-22-04324-f014:**
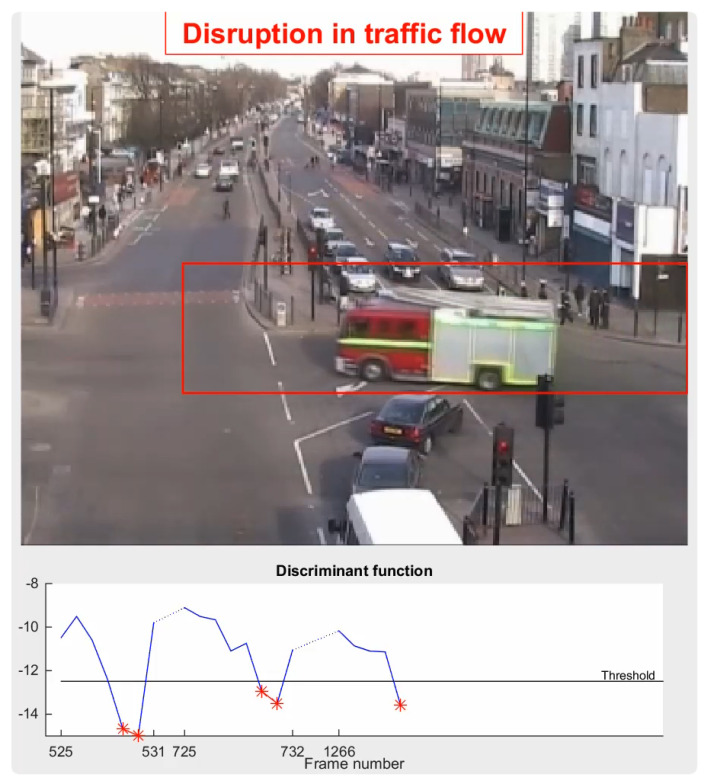
Multiple Object Detection with Topic Modelling [81]. The movement of the Fire engine, which is considered an abnormal event, in this case, is detected and marked with red boxes.

**Figure 15 sensors-22-04324-f015:**
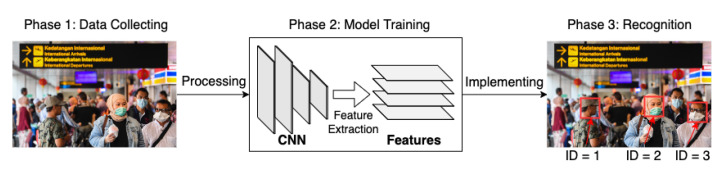
Block diagram of VFR. Images come from a public dataset Face Mask Detection [89].

**Figure 16 sensors-22-04324-f016:**
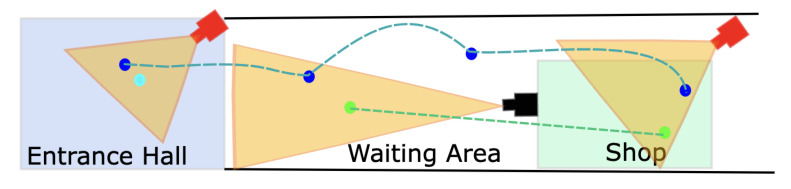
Person Re-Identification in video surveillance systems. This figure shows a top view of the region of interest comprising an entrance hall, a waiting area and a shop, monitored by three cameras (red is optical and black is thermal) with non-overlapping coverages (orange triangles). The locations of three individuals (blue, cyan and green dots) are shown at different timestamps in the region of interest. The person Re-ID aims to associate paths of the individuals denoted by blue and green dots detected by different modalities (optical and thermal) cameras.

**Figure 17 sensors-22-04324-f017:**
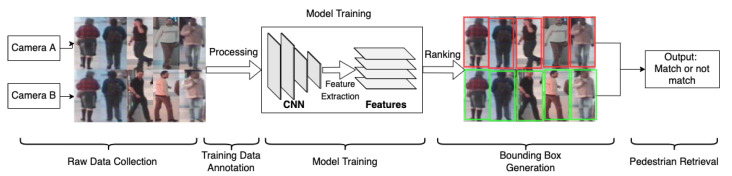
Block diagram of Person Re-ID. The images are from the CAVIAR dataset [78]. Five main steps for designing a Person Re-ID system: (1) raw data collection, (2) training data annotation, (3) model training, (4) bounding box generation and (5) pedestrian retrieval.

**Figure 18 sensors-22-04324-f018:**
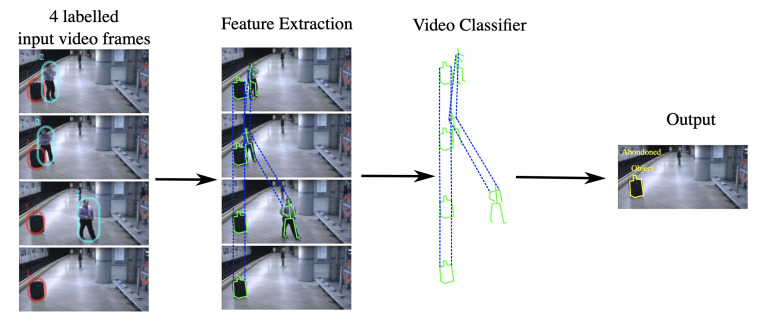
A block diagram of an HAR algorithm using an example of detecting an abandoned object. The images come from the i-LIDS bag and vehicle detection challenge of AVSS 2007 conference [97]. Four video frames with labelled objects are input to the HAR algorithm. The first step is feature extraction. The spatial and temporal information contained in the video helps in the recognition. Hence, both spatial (green) and temporal (blue) features are extracted. The video classifier classifies the extracted features. The surveillance operator is alerted if the video is classified as belonging to any of the pre-defined classes.

**Figure 19 sensors-22-04324-f019:**
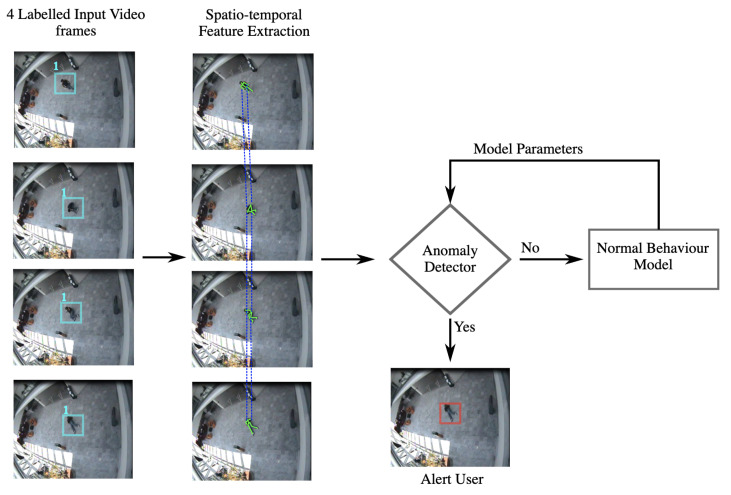
Block diagram of IUAD. The images come from the CAVIAR dataset [78]. Four labelled video frames are input to the algorithm. The spatial and temporal features are extracted first, followed by the anomaly detector. The anomaly detector uses parameters from a model trained on historical video data, which are all classified as normal. The output of anomaly detectors is either score or binary, where a binary output is used in this case.

**Figure 20 sensors-22-04324-f020:**
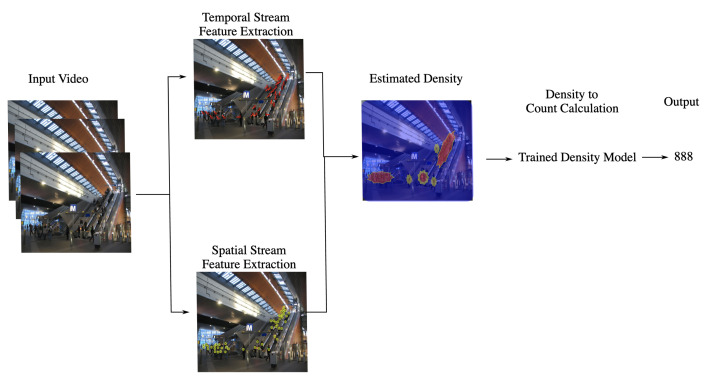
Block diagram of Crowd Count. The input frames are processed by two streams, spatial and a temporal, to extract features. The spatial features are represented by a yellow colour, and the temporal using a red colour indicates the direction of motion. The extracted features are processed to generate an estimated density map using different colours. The blue colour represents minimum or zero capacity, where yellow and red represent the bigger and maximum density. Finally, a well-trained density model is able to provide a prediction of the crowd number, which is 888 in this case.

**Figure 21 sensors-22-04324-f021:**
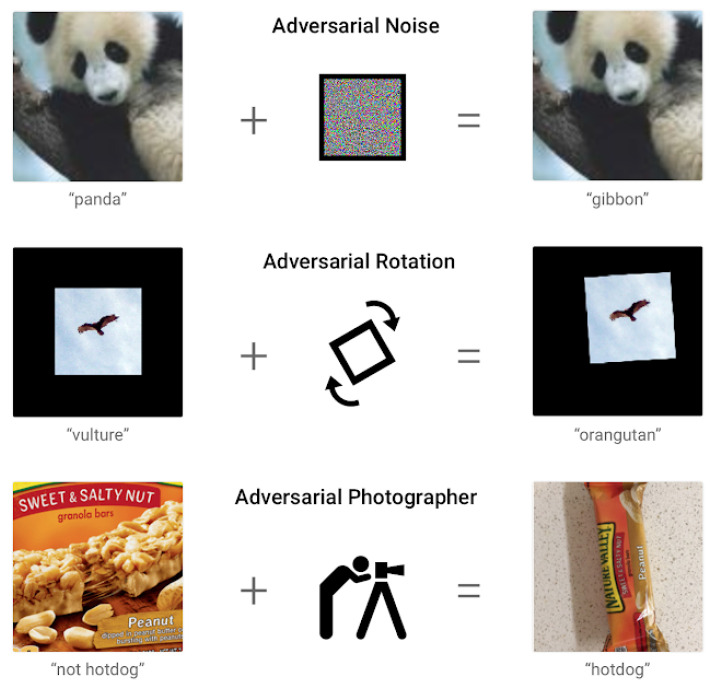
Adversarial attack examples. Image was taken from the Google AI Blog [127].

**Figure 22 sensors-22-04324-f022:**
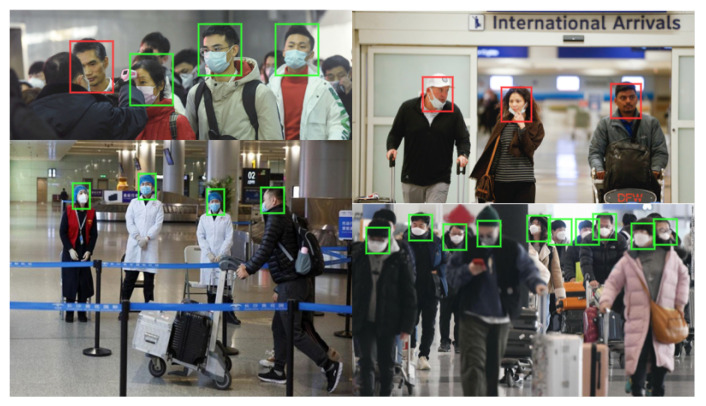
Facial recognition with masks. Images are from a public dataset Face Mask Detection [89].

**Figure 23 sensors-22-04324-f023:**
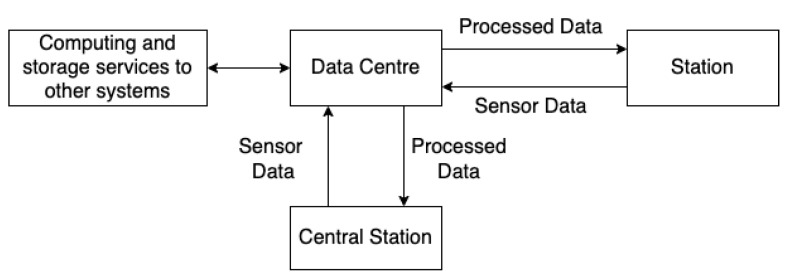
Data processing in the next generation of CCTV surveillance systems.

**Figure 24 sensors-22-04324-f024:**
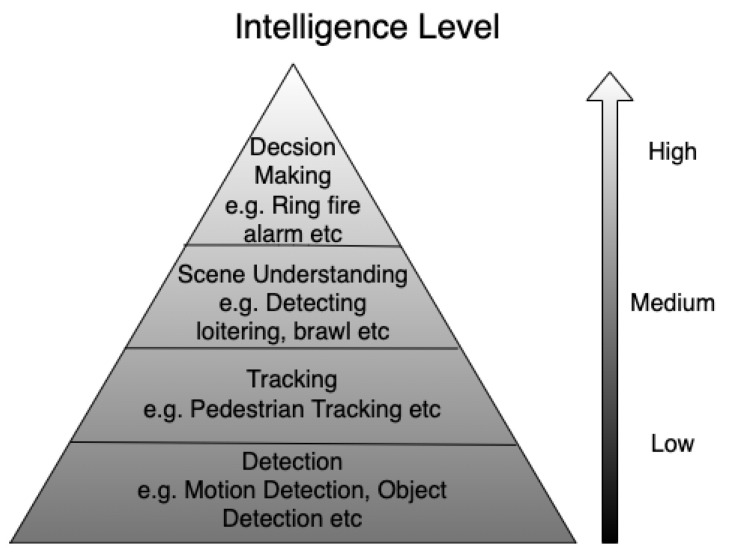
Intelligence levels of data analytics for CCTV.

**Figure 25 sensors-22-04324-f025:**
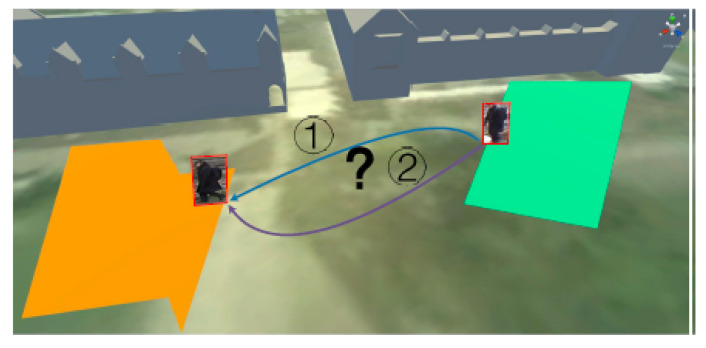
Event prediction in a blind zone [180]. This figure shows an example of two persons leaving the coverage of two different cameras (orange and green) and entering a blind zone. The video analytics system would predict and display the most probable trajectory of these individuals. The same concept could be used for the prediction of complex events such as criminal activity, possibly giving an operator extra time to respond.

**Figure 26 sensors-22-04324-f026:**
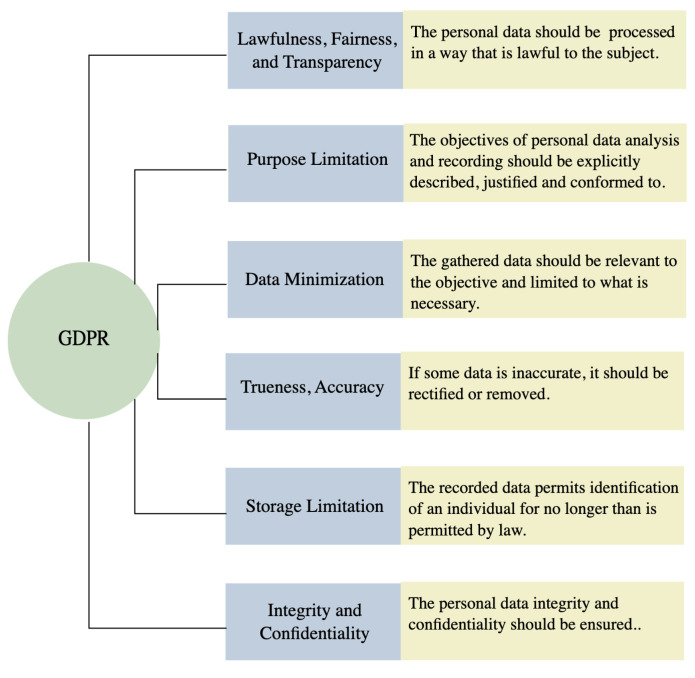
Six basic principles of GDPR.

**Table 1 sensors-22-04324-t001:** Number of publications of computer vision techniques used in rail networks surveillance by Google Scholar since 2010.

Computer Vision Techniques Used in Rail Networks Surveillance	Number of Publications Shown in Google Scholar
Motion or Moving Object Detection	83
Video Face Recognition	137
Person Re-Identification	6250
Human Activity Recognition	6160
Video Anomaly Detection	1060
Trajectory Analysis	1440
Crowd Analysis	1500

**Table 2 sensors-22-04324-t002:** Camera options (Basic).

Feature	Options	Pros/Cons
Connectivity Type	Analogue	Analogue point to point connection. Low-cost solution when upgrading conventional (analogue) CCTV system. Secure against cyber security threats.
	Digital	Digital (IP) interface. Flexible point to multipoint connection. High-cost solution when upgrading conventional CCTV system. Vulnerable to cyber security threats.
Connectivity Medium	Wired	Secure against cybersecurity intrusion. Less noise interference.
	Wireless	Flexible in terms of installation.
Field of view (FOV)	Fixed	Stationary mount. Monitor one small area of interest.
	Pan-tilt-zoom (PTZ) camera	Monitor large area using automatic/remote controlled pan (left/right), tilt (up/down) and zoom (in/out).
Lighting conditions	Day/night	Automatically adjust to light conditions. Coloured video during day and black and white at night. Used for conditions such as glare, direct sunlight, reflections and strong backlight.
	Low light	Used in low light, e.g., indoor restaurants, streetlights, etc. Cannot be used in completely dark conditions.
	Night vision/ Fog/Smoke	Used in complete darkness. Detect through obstructions such as fog and smoke. These are either active infrared (IR) or thermal cameras. The IR cameras use built-in IR illuminators and near IR (NIR) and IR cameras for monitoring. The thermal cameras are passive and might not provide video through glass or water. Thermal cameras are more costly than IR cameras.
Image sensor	Colour	Used in daylight or well-lit conditions. Provide accurate colours at the monitoring and recording systems.
	Monochrome	Used in near dark situations and give more details than a naked human eye can perceive. Might give low contrast during the day.
Housing	Dome	Spherical in shape to reduce wind and vibration effects. Protect and conceal camera direction. Some advanced units, called speed domes, rotate the camera and give an all around view.
	Sealed	Used in hostile situations. All electrical components are sealed to avoid explosion hazards.
	Impact resistant	Military grade. Used in high crime areas.
	Tamper resistant	Similar to impact-resistant but are additionally protected against tool vandalism. Usually resistant to cutting, hammering and prying.
	Bullet resistant	Similar to impact-resistant with additional safety for the glass.
Mount	Indoor	Wall. A bracket supports housing, and camera FOV is adjustable. Pendant. Suspend the camera from the ceiling. Corner. Used where two walls meet at the right angle (both interior and exterior). Dome. These are installed on the ceiling or other surfaces for dome type housings. These are susceptible to vibrations.
	Outdoor	Pole. These are used for unobstructed FOV. Corner. These are installed where two walls meet at the right angle. The best FOV is achieved by installing it close to the roof.
Image sensor	CCD	Stands for charged coupled devices. Used in daylight, lowlight and NIR cameras. Generate less heat. Susceptible to blooming (blooming is when a bright light source in the FOV hides some of the image details). Used for high resolution and quality images.
	CMOS	Stands for Complementary Metal Oxide Semiconductor. Low power and suitable for mobile devices or power constraint environments. Cheaper. Better video quality in outdoor areas on a bright sunny day.
Focal length (Lens)	Fixed	Used in fixed cameras for focusing on one area only.
	Varifocal	Focal length can be varied in a range, and focus is manually adjusted. Capture close-ups of activities at longer distances.
	Zoom	Used in PTZ cameras. Focal length can be varied in a range, but the focus is automatically adjusted.
Optical zoom factor		Optical zoom factor is calculated from the focal length range of the varifocal and zoom lens. For example, a range of 2–10 mm focal length represents a (10/2 =) 5× zoom factor. It does not mean 5 times enlargement of the image.
Aperture control	Fixed	Suitable for constant lighting conditions.
	Manual	Used for fixed cameras with controlled lighting. Less expensive as compared to the automatic. Requires a technician to operate.
	Automatic	Used in outdoor situations or where extreme changes in lighting conditions are expected.
Aperture size	Large	Used in dim light conditions. Image fore- and background are out of focus and blurred in daylight.
	Small	Complete scene will be in focus.
Filter	Neutral Density (ND)	Control the level of visible light and reduce it when it is too high.
	Polarising	Orient the light in a specific direction. Used to eliminate reflected light and glare. Improve image contrast.
	IR-cut	To control the NIR light sources such as the sun. Most image sensors are sensitive to NIR light. The NIR sources during the day can degrade the performance of image sensors. Without the IR-cut filter, the image will have unpredictable colours and poor contrast.

**Table 3 sensors-22-04324-t003:** Camera options (Value Added).

Options	Pros/Cons
External synchronization	The internal clock is synchronised to an external clock. It is important for any automation system, including CCTV, to have a synchronised clock.
Remote configuration	The cameras, especially the image properties, are remotely configurable.
Privacy masking	Selective blocking of private areas.
Covert cameras	These are hidden cameras and are preferably battery operated and wireless.
Dummy or drone cameras	These are non-functioning cameras and are installed for deterrence.
Auto scan of PTZ	Camera can perform automatic scanning of an area of interest.
Pre-set of PTZ	Camera orientation and lens setting can be programmed to scan/focus specific areas.
Slip ring PTZ	This allows the PTZ camera to rotate without twisting the cable. These are prone to contamination and temperature changes.
Motion detection	Camera can detect and provide alarm using built-in motion detection.
Backlight compensation (BLC)	The camera with BLC provides high-contrast images with a bright background.
Digital noise reduction (DNR)	The camera with DNR reduces the noise in the video. This noise is prominent in low-light or dark environments.
Mobile compatibility	The camera video can be viewed on a mobile device. The preferred choice is viewing without the installation of an additional application on the remote device.
Sun shields on housing	This protects the camera from direct sunlight, which can dramatically reduce the life of the camera.
Wipers on housing	These are similar to car windshield wipers. These are not recommended as they might erode the optical surface of the glass.
Heaters and ventilators for housing	Temperature differences between the interior and exterior of the housing can cause fog or icing on the glass. The heaters/ventilators help in maintaining the temperature inside the housing.
Smart camera	A smart camera is a small version of the CCTV system. It consists of a sensor, processor, video analytics and communication interface with a remote monitoring device.

**Table 4 sensors-22-04324-t004:** Comparison of displays.

Type of Display	Pros/Cons
Monochrome	Provides better details in some scenarios.
Plasma	Wider viewing angles. Higher contrast ratio. Higher black levels.
LCD	Compact and lightweight. Less power requirement. Low heat dissipation. Less prone to burn-in. No flicker. Long life. Low image contrast. True black colour is not produced. Restricted viewing angle.
LED	All benefits of the LCD. Better contrast. Wide colour range. Shorter life span. More expensive.

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
