# Peer review of "Recent Advances in Video Analytics for Rail Network Surveillance for Security, Trespass and Suicide Prevention—A Survey"

_sensors, 2022, doi:10.3390/s22124324_

Round 1

Reviewer 1 Report

This paper presents the recent development of   Video Analytics for Rail Network Surveillance.  State-of-the-art methods for object detection and behaviour recognition applied to rail network surveillance systems are introduced, and the ethics of handling personal data and use of automated systems are also considered. The paper is writen well. 

Here are some comments:

1, there are some pictures such as the ones fro the multiple object detection part which are not related to the railway background. It would be nice to replce them. 

2, It is not necessary to present the CNN and RNN network since this is ansurvey paper. 

3. The main chanlenges for the railway survailence can be pointed out for the readers. 

Author Response

We thank the Reviewer for these constructive comments helping us to improve the review paper. The introduced changes are marked ib blue.

Point 1: there are some pictures such as the ones from the multiple object detection part which are not related to the railway background. It would be nice to replace them.

Response 1: We have updated several figures including Figure 5, 6, 15, 17, 22 to make them more suitable to the railway background. Figure 13 and Figure 14 are from another case study of our work which is compatible with the algorithms stated in this paper in rail network systems. Since the results are quite illustrative, we would like to keep these images.  Also all figures that needed copyright approval by other journals were removed.

Point 2: It is not necessary to present the CNN and RNN network since this is an survey paper.

Response 2: Important computer vision methods applied on railway network surveillance such as object detection and facial recognition are described in the survey, some of which are deep learning methods, and include CNN and RNN frameworks. We agree that CNN and RNN networks are not necessarily introduced in detail since this is a survey paper. We have shortened the subsection 2.3.1. The revised Section 2.5.1 presents related work, and especially lines 273 to 296.

Point 3: The main challenges for the railway surveillance can be pointed out for the readers.

Response 3: We aree with the Reviewer. We have added a paragraph summarising the main challenges to railway surveillance, at the beginning of Section 2.3, lines 237 to 258 (in the revised version):

The main challenges to the railway network surveillance systems from computer vision perspective can be subdivided into two big groups depending on whether objects are moving or static:

1) classification, detection and segmentation and decision making for static objects such as left bags or a person laying on the rails of the network.

2) classification, detection and segmentation, tracking and decision making for dynamic (abnormal) events such as an unusual moving crowd, a terrorist leaving a bag on the platform, a theft running away or suicide of a passenger.

Both groups of approaches have in common that they need to be able to provide reliable solutions under a wide range of conditions, both naturally occurring (e.g. lighting changes, weather conditions changes and cyber attacks which include intentional interference during different stages of the imagery data transmission, processing and storage. Hence, there is a demand for a wide range of methods able to meet different requirements and provide trustworthy solutions. One way to provide a level of trustworthiness of machine learning approaches is to evaluate how the different above mentioned factors which we call uncertainties impact the developed solutions. Is the developed solution reliable and if yes, under what conditions and circumstances? The answer can be found based on probabilities or other information measures representing different levels of trust in the developed solution.

In addition, there are several other challenges which are mentioned: occlusions, environmental conditions, scalability - of the considered monitored area, big heterogeneous sensor data. In terms of network rail stations, we have small, medium and big train stations and in terms of rail networks for big geographic areas.

Reviewer 2 Report

  1. The area of “behaviour Surveillance” has not been covered well in the paper. There have been many states of the art publications in this context where the authors may get the benefit of looking at such as “Deep-Learning–Based App Sensitive behaviour Surveillance for Android Powered Cyber–Physical Systems”.
  2. How the ethics area has been discussed in the literature? This also needs further analysis. For example, how the CCTV systems is related to the ethics? This can be tightened in the paper.
  3. Although the authors have discussed the area from all angles, i.e., sensors, cameras…etc, the link among these angles are not yet clear in the paper.
  4. Some blocks of Fig7 are hard to read.
  5. What is the minimum number of frames are needed to detect the disruption in traffic flow? For example, in Fig13, the number of frames is varied.
  6. I feel there is a lot of repetition in the text. For example, Fig16 uses CNN, so does Fig8. Hence, you can see that the paper is verbose.
  7. You also need to add a statistical analysis of the existing work, not only summarising what they’ve done.

Author Response

We thank the Reviewer for these constructive comments helping us to improve this work. Below we summarise the main changes introduced in the revised paper.

Point 1: The area of “behaviour Surveillance” has not been covered well in the paper. There have been many states of the art publications in this context where the authors may get the benefit of looking at such as “Deep-Learning–Based App Sensitive behaviour Surveillance for Android Powered Cyber–Physical Systems”.

Response 1: This paper is mainly focused on the computer vision technologies perspective of railway surveillance. Indeed, there are other state-of-the-art deep learning based methods for pattern recognition and object/behaviour detecting tasks such as “Deep-Learning–Based App Sensitive behaviour Surveillance for Android Powered Cyber–Physical Systems”. However, due to the deployment difficulty and data privacy policy, these methods are not suitable for smart surveillance systems and video analytic. These methods can be seen as a modern type of data collecting methods instead of traditional camera sensors.

We added Section 5.3 “future sensors” to Section 2.3.1. We also added descriptions of these kinds of potential modern sensors for being used in surveillance systems in section 2.1, lines 155 to 158 (in the revised version).

Point 2: How the ethics area has been discussed in the literature? This also needs further analysis. For example, how the CCTV systems is related to the ethics? This can be tightened in the paper.

Response 2: We added a literature review of the ethics in public transportation systems surveillance at the beginning of section 6. The railway surveillance is implemented under the European Union General Data Protection Regulation (GDPR), e.g. lines 814 to 825 (in the revised version).

Point 3: Although the authors have discussed the area from all angles, i.e., sensors, cameras…etc, the link among these angles are not yet clear in the paper.

Response 3: The railway surveillance consists of three modules as shown in Figure 1: monitoring system, recording system and video analytic. The sensors make up the monitoring system. The monitoring and recording systems collect the video and other types of data and pass them to the video analytic module for further processing. We have added a bit of more description of the link among these angles in the caption of Figure 1 (now it is Figure 3 in the revised version) to make it clearer.  

Point 4: Some blocks of Fig7 are hard to read.

Response 4: Agreed. We have merged Figure 6&7 into one Figure (now it's Figure 8) to make it easier to read.  Please see Figure 8 (in the revised version).

In addition, several other Figures have been either removed or improved due to another reviewer’s comments and the copyright issues. These include Figure 5, 6, 15, 17, 22 (in the revised version).

Point 5: What is the minimum number of frames are needed to detect the disruption in traffic flow? For example, in Fig13, the number of frames is varied.

Response 5: In classical methods such as optical flow types and background subtraction, the minimum number of frames used to detect traffic flow disruptions is two, although the number of frames used can vary.

In more advanced machine learning approaches for disruption detection or anomaly detection, such as the example shown in Figure 13 (now it is Figure 14), the number of frames for the learning process varies, but the minimum number of  the frames required are two as well. A more specific description regarding this question has been added to section 3.1.2, lines 376 to 378 (in the revised version).

Point 6: I feel there is a lot of repetition in the text. For example, Fig16 uses CNN, so does Fig8. Hence, you can see that the paper is verbose.

Response 6: We agreed. Figure 16 aims to demonstrate how Person re-identification works based on deep learning algorithms. We have updated Figure 16 (now it is Figure 17) to make it clearer to avoid any repetition with other contexts.

Point 7: You also need to add a statistical analysis of the existing work, not only summarising what they’ve done.

Response 7: Thank you for this suggestion. We added a subsection “2.1. Bibliometric Analysis” with Figure 1, Figure 2 and Table 1 at the beginning of section 2, lines 74 to 104.

Round 2

Reviewer 2 Report

The authors have implemented significant improved on the paper with clear highlights on where these amendments are done. They responded to the comments in a sufficient way. I have no other comments.